# Importance of social inequalities to contact patterns, vaccine uptake, and epidemic dynamics

Adriana Manna [1], Júlia Koltai [2,3] & Márton Karsai [1,4] ✉

Individuals' socio-demographic and economic characteristics crucially shape the spread of an epidemic by largely determining the exposure level to the virus and the severity of the disease for those who got infected. While the complex interplay between individual characteristics and epidemic dynamics is widely recognised, traditional mathematical models often overlook these factors. In this study, we examine two important aspects of human behaviour relevant to epidemics: contact patterns and vaccination uptake. Using data collected during the COVID-19 pandemic in Hungary, we first identify the dimensions along which individuals exhibit the greatest variation in their contact patterns and vaccination uptake. We find that generally higher socio-economic groups of the population have a higher number of contacts and a higher vaccination uptake with respect to disadvantaged groups. Subsequently, we propose a data-driven epidemiological model that incorporates these behavioural differences. Finally, we apply our model to analyse the fourth wave of COVID-19 in Hungary, providing valuable insights into real-world scenarios. By bridging the gap between individual characteristics and epidemic spread, our research contributes to a more comprehensive understanding of disease dynamics and informs effective public health strategies.

Individuals' socio-demographic and economic characteristics are among the most significant factors that shape the dynamics of epidemic spreading processes. They not only influence the epidemic outcome in the hosting population but largely determine the course and severity of the disease for those who got infected[1]. There is a widespread agreement that pandemics disproportionately affect certain population groups rather than others[2–7]. Health-related inequalities in the burden of an epidemic partly arise from differences in the level of exposure to viruses and bacteria. These are associated with differences in social interactions, mobility patterns and work-related conditions, which are aggravated by disparities in the ability to be compliant with non-pharmaceutical interventions (NPIs), such as self-isolation, home-office and avoiding crowded places[8–13]. At the same time, inequalities in the severity and fatality of a disease can be accounted for by the heterogeneity in preexisting individual health conditions, protection attitudes and access to medical care, which are themselves related to socio-demographic and economic factors[14,15].

Although it is widely recognised that socio-economic inequalities play a crucial role in the transmission dynamics of diseases, traditional mathematical approaches have often overlooked these factors. Indeed, the state-of-the-art framework of modelling infectious diseases incorporates stratification of the population according to age groups[16] while discarding other potential relevant heterogeneities between groups of individuals belonging to different socio-economic strata. They commonly ignore the mechanisms through which these heterogeneities come into play, both directly and indirectly, in the

[1]Department of Network and Data Science, Central European University, Quellenstraße 51, Vienna 1100, Austria. [2]National Laboratory for Health Security, HUN-REN Centre for Social Sciences, Tóth Kálmán utca 4, Budapest 1097, Hungary. [3]Department of Social Research Methodology, Faculty of Social Sciences, Eötvös Loránd University, Pázmány Péter sétány 1/A, Budapest 1117, Hungary. [4]National Laboratory for Health Security, HUN-REN Rényi Institute of Mathematics, Reáltanoda utca 13-15, Budapest 1053, Hungary. ✉e-mail: karsaim@ceu.edu

different phases of an epidemic process. In traditional epidemiological models, contact patterns are usually represented in the aggregate form of an *age contact matrix* ($C_{ij}$), which encodes information on the average number of contacts that individuals of different age groups have with each other[17–22]. Moreover, not only the description of contact patterns is limited to an age structure, but also other epidemiological-relevant factors, such as vaccination uptake, infection fatality rates[23] or susceptibility[24], are usually described only by considering differences between age groups. While age is unarguably one of the most important determinants of these characters, the current literature falls short of understanding the role of other social, demographic, and economic factors in shaping human behaviour that are relevant to the epidemic's spreading. In recent years, researchers have advocated including social aspects in infectious disease modelling, arguing that the epidemic modelling community lacks a deep understanding of the mechanisms through which the socio-economic divide translates into heterogeneities in the spread of infectious diseases[25–28].

With this in mind, we aim to shed light on these mechanisms to address the following interrogatives: which are the most important individual characters and corresponding subgroups of the population that differentiate the most their epidemic-relevant behaviours, and how do these differences translate into epidemiological outcomes? We address these questions by analysing a large survey dataset coming from the MASZK study carried out in Hungary during the COVID-19 pandemic[29,30]. This data collects information on individuals' face-to-face interaction patterns in different contexts and other epidemiological-related behavioural patterns and opinions such as travel habits, vaccination uptake, or mask-wearing. The MASZK study consists of 26 cross-sectional representative surveys conducted monthly between 2020/04 and 2022/06 (for more details on the data, see Section 1 of SI and MM).

By considering the course of the pandemic in the country, we aggregate the data in six periods covering four epidemic waves (Ws) and two interim periods (IPs), as demonstrated in Fig. 1a.

Throughout this study, we are mainly interested in the dynamics and most influential determinants of social contacts that were recorded in the data as reported proxy interactions between pairs of individuals who spent at least 15 min within 2 m of each other on a given day. Outside of home, we distinguish between two contexts where social interactions may evolve. We differentiate between *work contacts* that emerge at the workplace (or at school) of respondents (or their minors) and *community contacts* that they evolved elsewhere than home or work. Meanwhile, we do not take into account household contacts in our study as we assume they do not change significantly during the different phases of the pandemic. Through the analysis of contact patterns, our aim is to show existing significant differences among subgroups of different socio-demographic characters, when accounting for the effect of age. Particularly, we demonstrate that dimensions such as employment situation and education level play a crucial role in determining contact numbers and vaccination uptake during a pandemic. Additionally, by proposing a new data-driven mathematical framework which explicitly considers further social dimensions other than age, we analyse the impact of such differences in terms of epidemic outcomes. Finally, by focusing on the Hungarian COVID-19 pandemic scenario, we reveal the unequal impact of the pandemic in terms of individuals belonging to different socio-economic statuses, where we differentiate individuals by their employment situation and income level. Note that although all the models have been completed on each pandemic period, for the demonstration of our findings, we exclusively show results about the 4th wave in the main text. We chose this period to demonstrate our

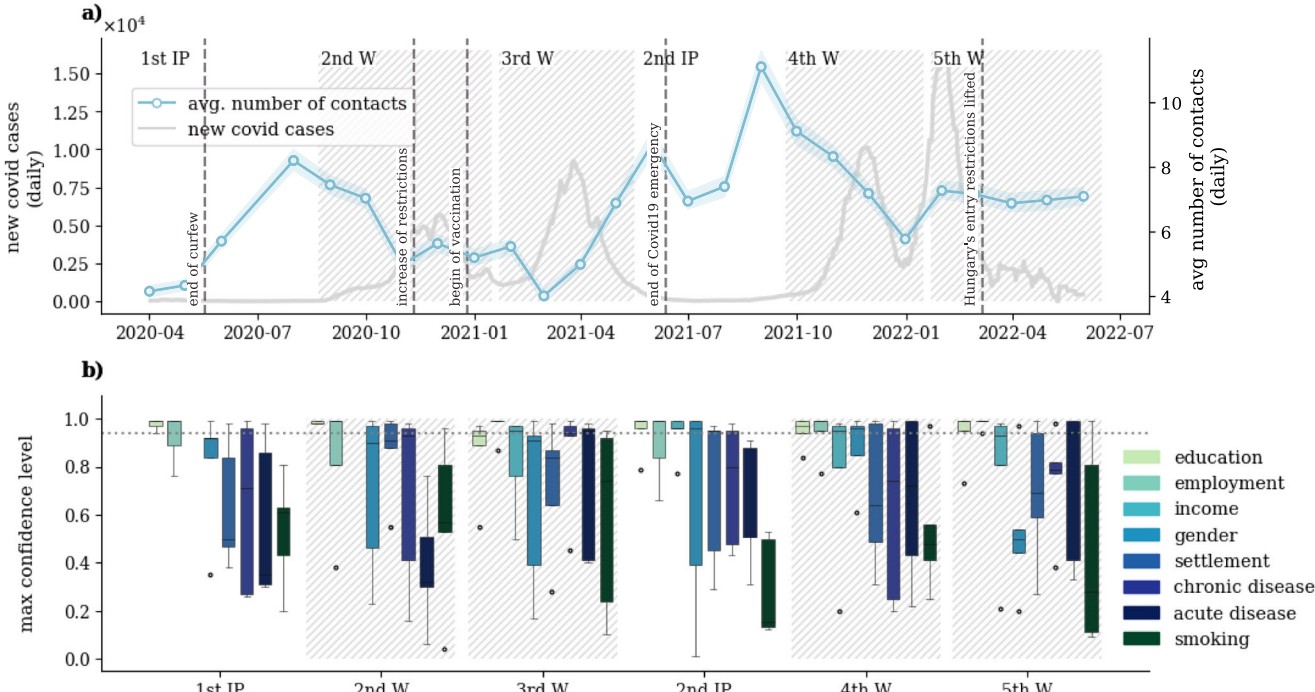

**Fig. 1 | COVID-19 trends in Hungary: cases, contacts, and key contact determinants. a** *Left axis*: number of new daily COVID-19 cases in Hungary from 2020/04 to 2022/07. *Right axis*: average number of daily contacts from 2020/04 to 2022/06, excluding household contacts. The values are shown as the median and interquartile range (IQR) of 1000 bootstrapped samples. The white and grey areas delimit the periods that have been aggregated in the analysis: two interim periods (IPs) (white areas) and four epidemic waves (W) (grey dashed areas). **b** Box-plot (outliers, minimum, lower quartile, median, upper quartile and maximum) of the maximum confidence level at which the effect of the different categories of the

variable on the total number of contacts becomes significantly different. The dispersion of the box plot refers to the variation of this value over different age groups. Results are shown for education level, employment situation, income level (which is present from the 2nd W, since this information was missing in the first few data collections), gender, settlement, chronic disease, acute disease, and smoking behaviour. The higher this value is, the more the variable influences people's number of contacts given their age. The horizontal dotted line is placed at 95% and it represents the confidence level at which the AME is considered to be statistically significant. Sample sizes are indicated in Section 1.2. of SI.

results because NPIs were not significantly changing, while the vaccination rate had saturated already during the 4th wave; this way, neither of these effects could bias the observed behaviours. We report our findings concerning other periods in the SI.

## Results

### The main determinants of human contact patterns

Human contact patterns represent the routes of infectious disease spreading by shaping the underlying transmission chain among susceptible individuals. During the COVID-19 pandemic, many aspects of human behaviour have experienced drastic interruptions in most countries worldwide. This was largely due to the implementation of non-pharmaceutical interventions (NPIs) that were installed to mitigate the spreading and other effects of the pandemic. They aimed at controlling the number of contacts, as well as influencing individual attitudes, to change the ways humans meet and interact with each other[24,31,32]. Their effects are evident in Fig. 1a, where the average number of daily contacts in Hungary is shown. This value increases during interim periods (IPs) when the numbers of daily infection cases are low and decreases during the epidemic waves (Ws) when infection risk is high, this way sensitively reflecting the adaptive behaviour of people throughout the pandemic.

Although at the aggregate level, these patterns are clear, there are non-trivial disparities at the level of individuals that may result in diverse contact patterns for given subgroups of the population. To explore these effects, in our statistical analysis, we focus on several

socio-economic dimensions that, interacting with age, may significantly affect the number of contacts that individuals have. We consider various socio-demographic variables such as individuals' education, employment, income, gender, settlement type, actual chronic or acute disease or smoking habits (for more details and definitions, see MM and Section 1.1 of SI). As a first observation, in Fig. 1b, we show the distribution of the *maximum confidence level* of the effects of these variables on the number of contacts in interaction with age during each period (for definition see MM). In these distributions, a higher value indicates higher certainty that a given variable has an effect, i.e., its effect is significantly different from zero at a smaller type *I* error probability[33], given the age of individuals. Based on these results, employment, education and income level were found to be the three most important dimensions in determining the number of contacts. This observation stands if we consider the overall number of contacts including both work and community relationships, and it is true as well if we only consider community contacts (with results shown in Section 2.1. of the SI along with all the robustness checks).

To further investigate the ways individuals of different characters adapt their number of contacts to the actual epidemiological situation, in Fig. 2 we show the average number of contacts over time decoupled by education level (Fig. 2a, c) and employment situation (Fig. 2b, d) for adult individuals older than 15 years old (for the corresponding plots related to *income* and *settlement* see Section 4.1 of SI, while for the plot decoupled by age groups see Section 4.2 of the SI). Results in panel (a) suggest that high and mid-high-educated individuals have consistently

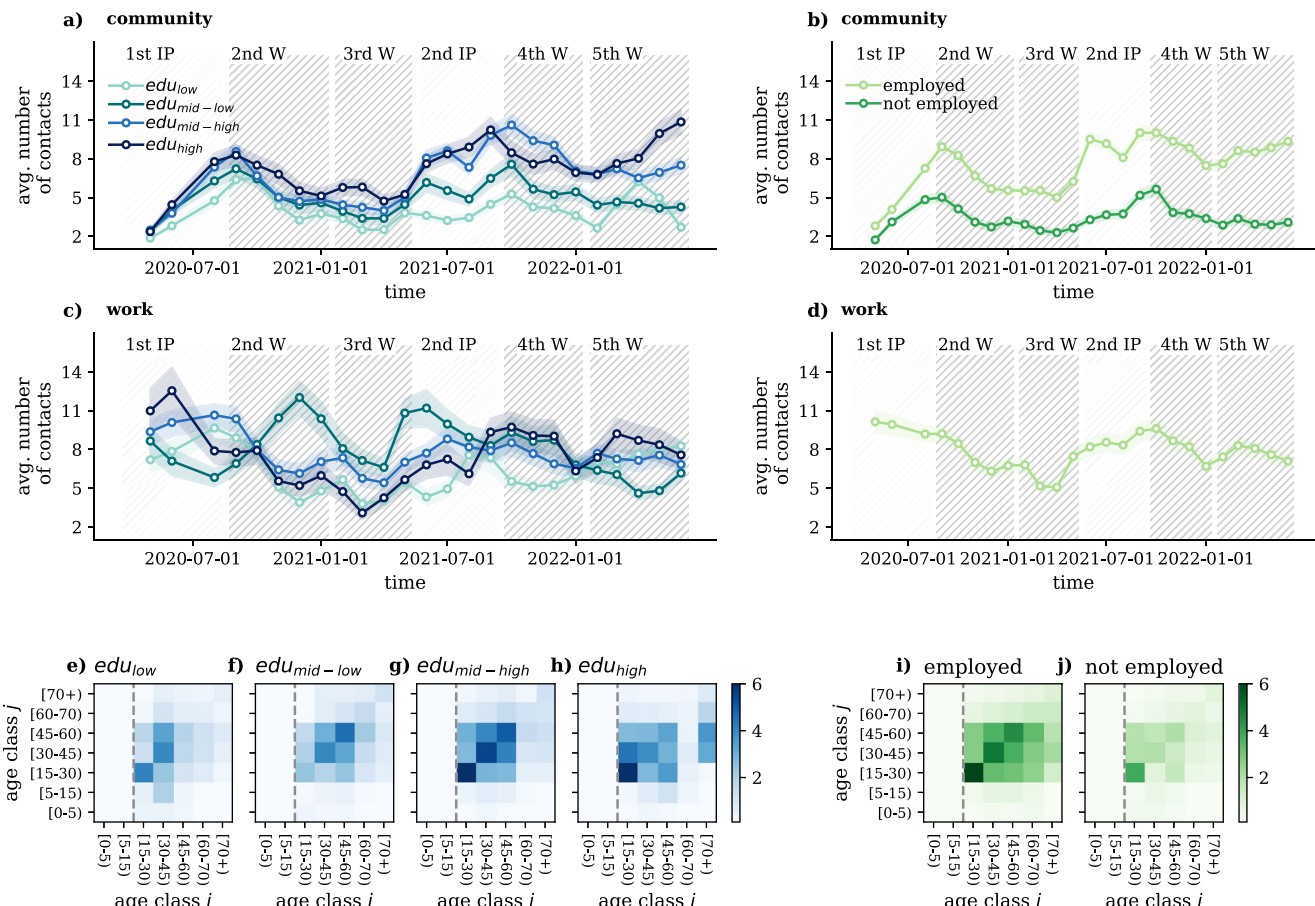

**Fig. 2 | Contact dynamics by population subgroups. a** and **b** Average number of contacts in the community layer **a** by different education levels and **b** employment situation. **c** and **d** Average number of contacts at workplace **c** by different education levels and **d** for employed people. All curves have been smoothed over the observation periods for better visualisation. **e–h** Decoupled age contact matrices by education level for the 4th epidemic wave. **i** and **j** Decoupled age contact matrices by employment situation for the 4th wave. These figures depict contact numbers only for the adult population [15+], while matrices containing children are shown in Section 4.3.2 of the SI. All the values are shown as the median and IQR of 1000 bootstrapped samples.

higher number of contacts in the community layer throughout the observed period. In addition, the pattern of the average number of contacts of these groups suggests that they were probably able to better adapt to the epidemiological situation and NPIs by decreasing their contacts number during epidemic waves and increasing again during interim periods. Conversely, individuals with mid-low and low education levels exhibit a lower and relatively stable number of contacts over time. Workplace contact dynamics suggest that only highly educated individuals (high and mid-high) were able to adapt to the epidemiological situation, while seemingly, those with lower education levels had less flexibility to adjust during different pandemic periods. Interestingly, mid-low educated individuals reported a higher number of workplace contacts, particularly during the second wave (2nd W) and the second interim period (2nd IP), attributable to their predominant vocational degree status and involvement in on-site interactive work. These patterns are confirmed when examining the relative number of contacts over time, as elaborated in Section 4.1.1 of the SI.

When we group people by their employment situation, it makes sense to compare groups in the community layer. From Fig. 2b it is clear that employed people maintain more contacts even outside of their workplace as compared to not-employed individuals, which is a clear sign of behavioural differences between these two groups. Meanwhile, employed individuals follow somewhat similar contact dynamics as highly educated people (see Fig. 2d), signalling some correlation between these two groups.

From an epidemic modelling perspective, the most convenient way to code interaction patterns between different groups is via contact matrices that quantify the average number of interactions between population strata. Contact matrices allow models to depart from the homogeneous mixing assumption, i.e. taking all individuals to meet with the same probability. Instead, they allow the introduction of non-homogeneous mixing patterns between different groups, while keeping the model computationally more feasible as compared to contact network-based simulations. Conventionally, epidemic models incorporate $C_{i,j}$ age contact matrices that code the average number of contacts between people from different age groups (for formal definition, see MM). Nevertheless, age contact matrices could be further stratified by other socio-demographic characters that influence the contact numbers of individuals. In Fig. 2e–j, we show the age contact matrices decoupled by education level and employment situation ($C_{d,i,j}$) for the 4th epidemic wave for the adult population (see MM for more details and Section 4.3.2 of SI for the corresponding matrices including children). These matrices have been computed by considering *community*, *work* and *household* contacts together. The emerging large differences between these matrices demonstrate clearly that beyond age, the identified variables, i.e. education and employment status, induce significant differences in the contact patterns of individuals. Although these variables may not be independent of the age of people, the observed distinct patterns suggest more complex mechanisms controlling contact patterns among subgroups that cannot be explained by age alone.

**Beyond age stratification.** We demonstrate that social inequalities significantly influence human contact patterns, thereby shaping the network of proxy social interactions. This is critically important for the propagation of diseases as it determines the transmission chain of an infection spreading among a susceptible population. Consequently, incorporating the contact pattern differences among individuals of different socio-economic backgrounds into epidemiological models is crucial. This could help to understand the unequal spread and uneven burden that an epidemic could impose on the different socio-demographic groups of a society. To this end, we propose a simple mathematical framework based on the extension of a conventional age-structured SEIR compartmental model[34,35], which we call the *extended SEIR* model. The *conventional SEIR* model assumes that each

individual in a population is in one of the mutually exclusive states of Susceptible (S), Exposed (E), Infected (I) or Recovered (R). Transitions of an individual between these states are controlled by rates (S$\longrightarrow^{\lambda}$E, E$\longrightarrow^{\varepsilon}$I, I$\longrightarrow^{\mu}$R) with the $\lambda$ rate influenced by the frequencies of interactions between age groups coded in a $C_{i,j}$ age-contact matrix. The proposed extended model incorporates $C_{\bar{d},i,j}$ age contact matrices instead, that are decoupled along important socio-demographic dimensions $\bar{d}$ to model epidemic spreading in different subgroups of the population (see MM and Section 6 of the SI for further details).

Particularly, we analyse the impact of decoupled age-contact matrices along four dimensions: employment situation, education level, settlement, and income level (for exact definitions and possible variable values, see MM). Taking the decoupled contact matrices as input, we simulate the spread of infectious disease among an entirely susceptible population using both the *conventional SEIR* and the *extended SEIR* models. Having fixed the epidemiological parameters such as the transmission rates and seeding strategy, other input parameters like the population distributions and contact matrices have been estimated from data, as we explain in Section 7 of the SI in more detail.

The proposed model allows us to investigate how differences in contact patterns along diverse social groups translate into an unequal burden of the epidemic. To quantify these differences in the epidemic outcome, we measure the attack rate defined as the population-wise normalised fraction of individuals who contracted the infection from a given group. To follow the distribution of the people along the investigated dimensions, we show the survey population fractions in the different age groups in Fig. 3a–d. Meanwhile, in Fig. 3e–h, we depict the attack rates calculated using the *extended SEIR* models for different age and socio-demographic groups (and as reference only for age—see grey solid lines). Results are shown for the cases when we decouple each age group along the four dimensions analysed. As anticipated by the statistical analysis, employment and education produce the largest differences between groups in terms of attack rate by age. Interestingly, the group of employed people happened to be the most infected group in all age groups, while mid and high-educated individuals are more infected among those who are 45–60 years old. When decoupling age contact matrices by settlement and income, although differences appear smaller between groups, high-income individuals and the ones living in the capital are more infected, particularly elderly ones with age 60+. These modelling results suggest the interesting overall conclusion that employed and wealthier group of the population, as well as those living in the capital report a higher attack rate, thus they are typically the most infected group relative to their population size.

These results also demonstrate that the *extended SEIR* model is able to capture differences introduced by the considered socio-demographic variable and, in this way, to model the epidemic impact on the different groups of the population. These differences are also visible at the population level. In Fig. 3i, we show the differences between the attack rates predicted by the *conventional* and the *extended SEIR* models for each age group and overall, too. It is evident from these results that models using contacts only stratified by age may overestimate (negative difference) the size of the epidemic in different age groups or in the whole population. For example, our simulations based on data from the 4th wave demonstrate that the *conventional SEIR* model could predict higher attack rates for each age group with respect to the *extended SEIR*, which considers differences in contact numbers along the employment situation or the education level. (See Section 7.1 of SI for the corresponding figures for the other periods and the relatively sensitive analysis). It is important to highlight that the uneven age distribution within the different subgroups sometimes reduces or annul the effect of the difference in the contact patterns when we are computing aggregate quantities at the population level. This explains why, even if there is a significant difference in contact patterns, the difference in the overall attack rates only spans a small range between the two models.

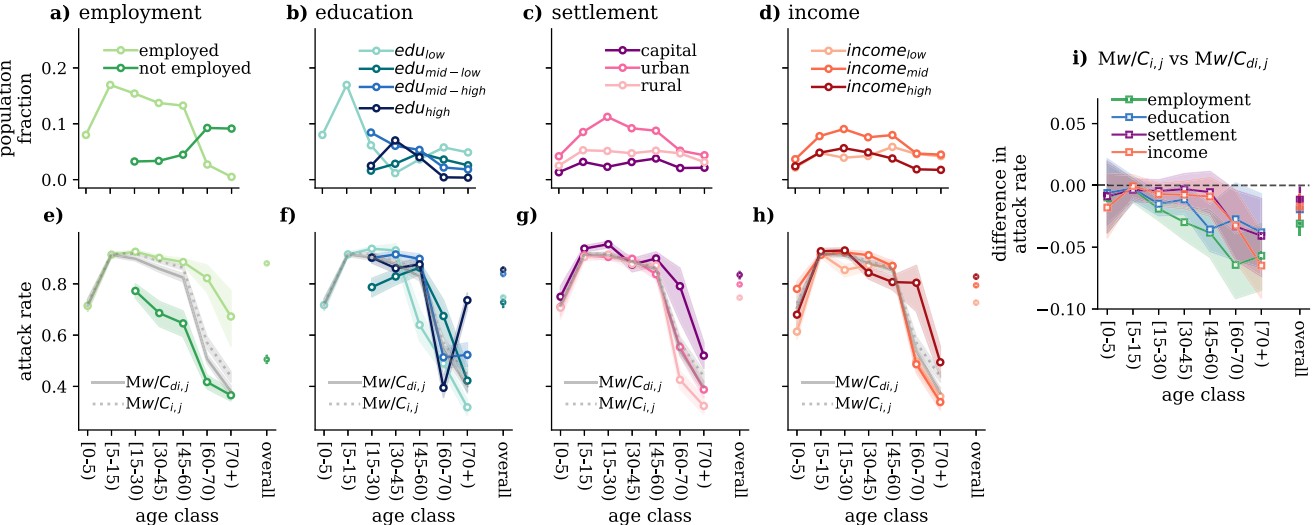

**Fig. 3 | Differences in attack rates in extended SEIR models. a–d** Survey population distribution by age and employment situation (**a**), education level (**b**), settlement (**c**) and income (**d**). **e–h** Attack rate by age and employment situation (**e**), education level (**f**), settlement (**g**) and income (**h**) as predicted by the *extended SEIR* (i.e. calculated with both age and the given socio-economic variable stratification). The grey lines represent the attack rate by age calculated only with age stratification as predicted by *classical SEIR with* $C_{i,j}$ (solid lines) and *extended* *SEIR with* $C_{di,j}$ (dotted lines). **i** Difference in the attack rate by age as predicted by the *classical* ($Mw/C_{i,j}$) and the *extended* ($Mw/C_{di,j}$) model when different, the dimensions are considered. Results are shown for the 4th wave. Epidemiological parameters: $\mu = 0.4$, $\epsilon = 0.25$, and $R_0 = 2.5$. Simulations start with $I_0 = 5$ initial infectious seeds. Results are sown as the median and IQR computed over 1000 simulations.

## Vaccination uptake and contact number differences

Beyond the crucial role played by the network of face-to-face interactions, individual vaccination uptake may also substantially affect epidemiological outcomes by decreasing morbidity and mortality. By applying the same pipeline of statistical analysis as we explained above, we identify the dimensions along which individuals made different decisions in terms of vaccination, given their age. In this case, the interaction with age is particularly important given that the COVID-19 immunisation strategy implemented in Hungary followed an age-stratified outreach by prioritising elderly individuals[36,37]. Despite the prioritisation of specific groups, such as medical personnel, in the initial months of the vaccination campaign, we regard the associated impact as negligible for the purposes of this study. Interestingly, the statistical analysis in this case indicates income as the most important dimension along which individuals made different vaccination decisions (see Section 2.2 of SI for the results of the statistical analysis). Figure 4a–d shows the percentage of vaccinated individuals by age and the investigated dimensions during the 4th wave of the pandemic. Although by the 4th epidemic wave the vaccination saturated in Hungary, the effects of the age-dependent vaccination policy are clearly visible. More strikingly, we find that higher socio-economic groups of the population were more likely to get vaccinated against the COVID-19 virus. This observation is valid for all age groups and periods considered in the analysis (see Section 5 of SI for the corresponding figures for the other periods).

To consider these observations, we model the vaccination uptake in the *extendend SEIR* framework. More precisely, we define the probability of getting vaccinated (i.e. immune or recovered from the point of the infection) to be dependent, in this case, on both the age and the subgroup of the population considered. Using this *extended SEIR* model, we are able to compare the effects of vaccination uptake, while keeping fixed the structure of contacts. Figure 4e–h shows the *averted attack rate* due to vaccination with respect to the non-vaccination scenario. We consider the probability of getting vaccinated along the four different investigated dimensions separately. In all of these scenarios, the gain in averted infection is strongly dependent on the subgroup membership. As expected, the groups with higher vaccination uptake are the ones, which reduce their attack rate

the most in the vaccination scenario. However, this pattern is not linear. For example, among individuals aged 60+, although the not-employed people report a higher vaccination uptake, they are the ones that gain less in terms of averted infections. This is because these individuals, having a low number of contacts, are already protected from exposure to the virus; thus, they gain less from vaccination (see Section 7.2 of SI for the corresponding figures for the other periods and the relative sensitive analysis).

## Stratified modelling of the Hungarian scenario

To provide an example of how the proposed mathematical framework can be applied to a real case scenario, we model the 4th COVID-19 wave in Hungary between 09/2021 and 01/2022. As the statistical analysis showed that employment and income are the most important dimensions along which, respectively, contact patterns and vaccination uptake change the most, here we divide the population into subgroups by considering simultaneously these two additional dimensions other than age. In addition, we introduce a new compartment $D$ to our SEIR model, which represents a dead state that infected individuals may enter with a transmission rate $I \xrightarrow{\mu^{IFR}} D$.

To simulate the SEIRD for the 4th COVID-19 wave in Hungary, we calibrate our model using the Approximate Bayesian Computation (ABC) method[38,39] on the total number of daily deaths from 09/2021 to 01/2022[40]. Details about the fitting method and calibrated results are summarised in Section 8 of the SI.

The results of the simulated model are presented in Fig. 5, which shows the daily fraction of newly infected (panels (a)–(c)) and new dead (panels (d)–(f)) cases for different employment, income, and age groups. As expected, these curves suggest that the group of employed people experienced the infection at a higher rate as compared to those not employed. At the same time, in terms of socio-economic status and age, more affluent and younger people got infected more during the simulated epidemic wave. On the other hand, strikingly the contrary trend is suggested in terms of mortality rates. From the simulations, we find that although not-employed, low-income and older individuals appeared with the lowest infection rates, they evolved with the highest mortality rate as compared to other groups.

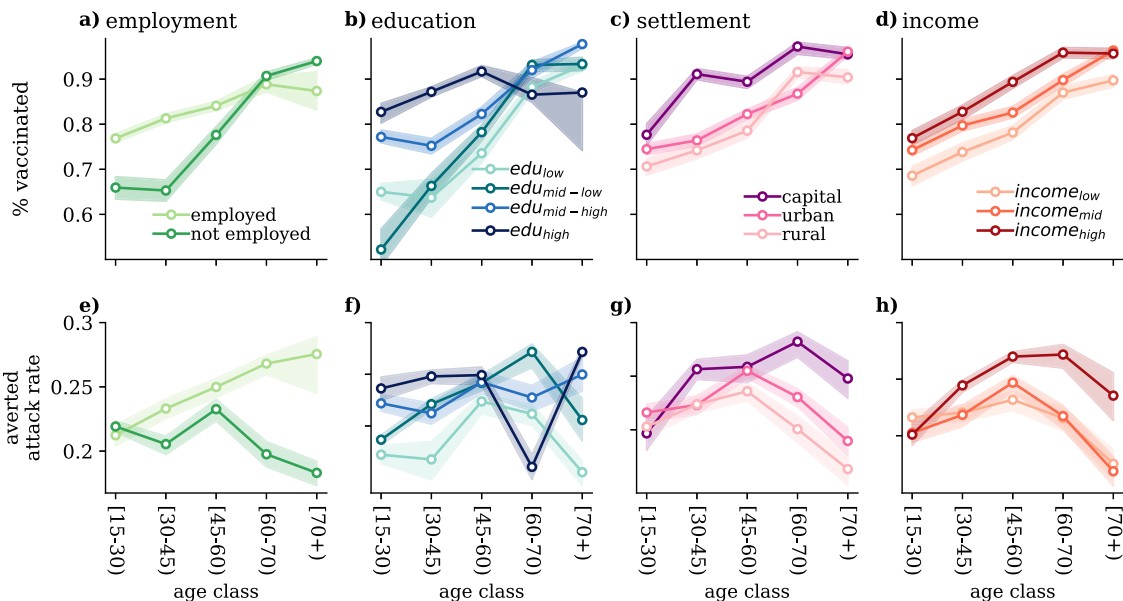

**Fig. 4 | Disparities in COVID-19 vaccination and averted attack rates. a–d** Fraction of individuals vaccinated with at least one dose against COVID-19 during the 4th wave, decoupled by age and **a** employment situation, **b** education level, **c** settlement, and **d** income. Panels **e–h** show the averted attack rates (difference between the attack rate in the non-vaccination scenario with respect to the vaccination one) by **e** age and employment situation, **f** education level, **g** settlement, and **h** income as predicted by *extended SEIR*. The model takes into account different rates of vaccination uptake by subgroups of the given variable compared to the non-vaccination scenario. Epidemiological parameters: $\mu = 0.4$, $\epsilon = 0.25$, and $R_0 = 2.5$. Simulations start with $I_0 = 5$ initial infectious seeds. Results are sown as the median and IQR computed over 1000 simulations.

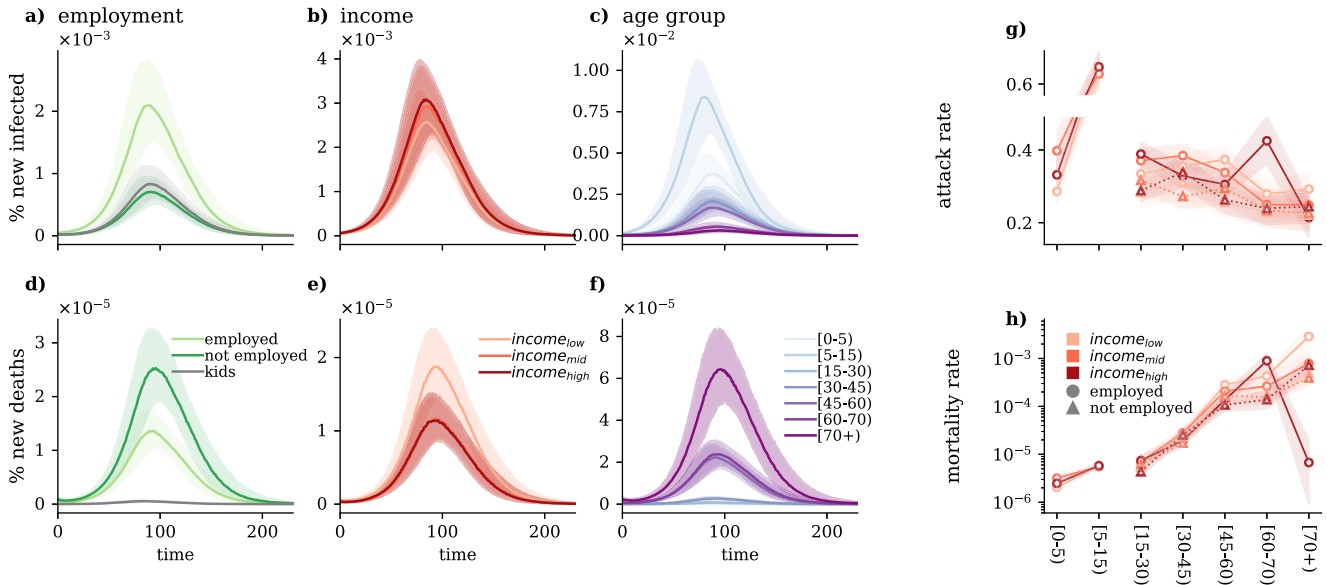

**Fig. 5 | Modelled epidemic dynamics in Hungary during the 4th COVID-19 epidemic wave.** Results from the simulated model for the 4th wave in Hungary. The figure shows the median and the IQR over 1000 runs. **a–c** Fraction of newly infected by **a** employment situation, **b** income level, and **c** age group. **d–f** Fraction of new deaths by **d** employment situation, **f** income level, and **g** age groups. **g** Attack rate by age, employment situation and income level. **h** Mortality rate by age, employment situation and education level. For more details about the parameters of the numerical simulations, see SI.

Considering that the COVID-19 fatality rate of infected individuals (IFR) depends on their age, this observation can be largely attributed to the fact that not-employed and low-income individuals are also the oldest ones in the population. To explicitly separate the effect of age when it comes to analysing the burden of the epidemic, we examine the attack rates and mortality rates by age groups separately in each of the subgroups of the population stratified by income level and employment status (Fig. 5g, h).

As we have shown in the analysis above, these results confirm an overall decreasing infection rate by age and that not-employed individuals experience the lowest attack rate in each age group. The further stratification of not-employed individuals by income level reveals a clear pattern, with high-income people exhibiting a higher infection rate compared to mid-income and low-income individuals. In contrast, the infection pattern among income levels of employed individuals is age-dependent. For young employed individuals in the age group

15–30, high and mid-income individuals register a higher share of infections, while among those older than 30, low-income individuals exhibit a higher infection rate compared to mid and high-income individuals. An exception is represented by the age group 60–70, for which the most infected group is still the high income.

In terms of mortality, as expected, we find an increasing trend by age; otherwise, we can conclude a similar pattern as for the attack rate. Particularly, the mortality rate by age group shows that employed people die at a higher rate. While in terms of income, other than the group aged 15–30 and 60–70, the lower-income people suffered more deaths, according to our simulations. The extreme decrease in mortality rate for the employed high-income group is due to data sparsity in the survey data, recording only a few data points in this category (see Section 1.2. of SI).

## Discussion

Several factors may determine how an infection would turn out for a given person. Some of them are coded genetically or determined by physiological conditions, but many of them are environmental and correlate with one's socio-demographic characteristics. In any society, people show uneven patterns along numerous social, demographic, and economic characteristics, like age, income or employment status. These characteristics not only induce medical disparities between people (as in immunity, overall health conditions, or chronic diseases) but naturally translate to differences in adaption capacities and other behavioural patterns, allowing certain groups to be more exposed to infection. The simultaneous actions of all these factors lead to observable inequalities in terms of epidemic burden between different groups at the population level.

This study highlights the significant impact that social determinants have on human behaviours that are relevant to epidemic transmission. Specifically, exploiting the data of the *MASZK* study[29,30], we show that contact patterns and vaccination uptake are influenced by socio-economic factors. Our findings suggest that contact patterns are shaped by social factors not only in their absolute values but also in the extent to which they vary in response to extraordinary events, such as lockdown or curfew interventions. Specifically, our statistical analysis shows that socio-economic factors such as employment situation and education level played a significant role in determining contact numbers and vaccination uptake during the COVID-19 pandemic in Hungary. Additionally, in contrast to studies reporting a negative correlation between poverty and the number of contacts[41], we find that people with higher socio-economic status tend to have a higher number of contacts and are the ones that change the most their number of interactions with respect to the epidemiological situation. Contrarily, people with lower socio-economic status maintain a lower number of contacts over time with smaller oscillations. Although we cannot establish a causal connection as a possible and plausible explanation these results suggest that people from higher socio-economic groups, such as those with higher education, income, and employment status, were able to better adapt to the epidemiological situation and NPIs and were more likely to get vaccinated.

We propose a mathematical framework that extends the well-known age-stratified approach to model infectious diseases by explicitly accounting for differences in contact patterns and vaccination uptake for specific subgroups of the population. This method allows us to better understand the mechanisms underlying the emergence of inequalities in epidemiological outcomes. Results demonstrate that traditional epidemiological models, that only consider age, could overlook crucial heterogeneities along other social and demographic aspects that may impact the spreading of an epidemic. Through simulated epidemic processes, we show that significant differences in terms of attack rates arise from differences in contact patterns. By neglecting differences in vaccination uptake and the effects of vaccination

campaigns among subgroups in modelling, we would miss important determinants which significantly influence the outcome of an epidemic.

By simulating a pandemic period in Hungary, we reveal the unequal health-related impact of the COVID-19 pandemic among individuals belonging to different socio-economic groups. Although the higher number of contacts translates into higher attack rates for wealthier and employed individuals, the age structure and the vaccination decision of such groups translate into lower mortality rates for these individuals, while disadvantaged groups are the ones suffering higher mortality. These results are in line with the empirical findings of refs. [42,43] for the 2nd and 3rd COVID-19 waves in Hungary. In those studies, the authors find that individuals living in more deprecated areas are associated with a lower risk of being ascertained as a confirmed COVID-19 case and a higher risk of death. Additionally, during the 3rd wave, those were associated as well with lower vaccination uptake. However, we recognise as a limitation of our work that, owing to data gaps, the initialisation of population distributions within compartments and socio-demographic groups for the calibrated model is done only proportionally the population distribution. Indeed, the assumption of a homogeneous initial condition is improbable precisely because of the different contact patterns, exposure risks, and vaccination uptake among different age and socio-economic groups (see Section 8 of SI for further detail).

Due to the limitation of the survey collection methodology, contact patterns of individuals can be differentiated only by the characteristics of participants. Indeed, the only information we know about the contacted peers is their age, while their other characteristics remain unknown. Thus, our extended *SEIR* model can only account for age-contact matrices that are decoupled along other social dimensions of the participants (ego). In other words, although our model incorporates additional social dimensions, given the subgroup the *ego* belongs to, it still only considers the average number of contacts stratified by the age group of the contacted (alter). In order to introduce a *generalised contact matrix*[44] stratified along multiple socio-demographic dimensions of the contactee, we would need information about such dimensions. Such information can be collected via detailed contact diaries[30], which are based on the reports of the respondents about peers and commonly suffer from recall bias and other limitations[45,46]. Moreover, we acknowledge the interplay between vaccination and contacts in mutually shaping each other, yet we have opted not to delve deeper into these mechanisms in the current work as they may fall beyond the scope of our study.

By shedding light on the complex interplay between social, demographic and economic factors and disease transmission dynamics, our findings underline the need for a new mathematical framework for epidemic modelling that accounts for multidimensional inequalities. This would help us to better understand the socially stratified consequences of an epidemic and highlight non-negligible inequalities between different socio-demographic groups. Additionally, incorporating social factors into epidemiological models will provide a valuable tool to design and evaluate targeted NPIs to cope more efficiently with the spread of an infectious disease.

## Methods
### Data description
The data used in this study comes from the MASZK survey[29,30], a large data collection effort on social mixing patterns made during the COVID-19 pandemic. It was carried out in Hungary from April 2020 to July 2022 on a monthly basis. The data was collected via cross-sectional anonymous representative phone surveys using the Computer Assisted Telephone Interviewing (CATI) methodology and involved a 1000 large nationally representative sample each month. During the data collection, participants were not asked for

information that could be used for their re-identification. The phone survey data collection was carried out after informed consent from the respondent at the beginning of each interview. Any information about children was obtained by asking questions from their parents or legal guardians after consent. Survey data was not collected from any under-age subject. The data collection fully complied with the actual European and Hungarian privacy data regulations and was approved by the Hungarian National Authority for Data Protection and Freedom of Information[47] and also by the Health Science Council Scientific and Research Ethics Committee (resolution number IV/3073-1/2021/EKU).

The primary goal of the data collection effort was to follow how people changed their social contact patterns during the different intervention periods of the pandemic. Relevant to this study, the questionnaires recorded information about the *proxy social contacts*, defined as interactions where the respondent and a peer stayed within 2 m for more than 15 min[48], at least one of them not wearing a mask. Approximate contact numbers were recorded between the respondents and their peers from different age groups of 0–4, 5–14, 15–29, 30–44, 45–59, 60–69, 70–79, and 80+. Of which we aggregate the last two groups in 70+. Contacts number data about underage children were collected by asking legal guardians to estimate daily contact patterns.

Beyond information on contacts before and during the pandemic, the MASZK dataset provided us with an extensive set of information on *socio-demographic characteristics* (gender, education level, etc.), *health conditions* (chronic and acute illness, etc.), *financial and working situation* (income, employment status, home office, etc.), and *attitude towards COVID-19 related measures and recommendations* (attitude towards vaccination, mask-wearing, etc.) of the participants. In order to study different stages of the pandemic, we consider six epidemiological periods, including three epidemic waves (Ws) and three interim periods (IPs) (see Fig. 1a).

On the collected data, a multi-step, proportionally stratified, probabilistic sampling procedure was elaborated and implemented by the survey research company using a database that contained both landline and mobile phone numbers. The survey response rate was 49%, which is notably higher than the average response rate (between 15% and 20%) of telephone surveys in Hungary. The sample is representative of the Hungarian population aged 18 or older by gender, age, education and domicile. To correct sampling biases, we used individual weighting to decrease the difference between population and sample distribution of social-demographic variables. The weights were calculated by the survey research company responsible for the data collection. For the calculation of the weights, raking was used[49], which relies on iterative proportional fitting[50]. More details about the weighting procedure can be found in Section 1.3 of SI. After data collection, only the anonymised and hashed data were shared with people involved in the project after signing non-disclosure agreements.

## Sociodemographic dimensions

The sociodemographic dimensions that we analyse are the following: (i) *education level*, which can have four possible levels: low, mid-low, mid-high and high; (ii) *employment situation*, which can be either employed or not-employed, including students and retirees individuals; (iii) *perceived income* (called simply *income* through the manuscript) can have three possible levels: low, mid and high; (iv) *gender* refers to the biological gender and can be either female or male; (v) *settlement*, which refers to the area where individuals live and can be either capital, rural or urban; (vi) *chronic disease* is a Boolean dimension indicating if an individual is affected by any chronic disease; (vii) *acute disease* is a Boolean dimension indicating if an individual is affected by any acute disease, and (viii) smoking is a Boolean dimension indicating if an individual is a smoker or not. A detailed explanation of these variables is provided in Section 1.1 of the SI.

## Data prepossessing

All the analyses on the number of contacts have been performed after having deleted the outliers at the 98% percentile with respect to the period of interest. All the results presented in this work have been computed by accounting for each participant according to its representative weight, as detailed in Section 1.3 of SI. In addition, to assess the uncertainty of the estimation of contacts and contact matrices, we employ the bootstrapping sampling technique, as described in Section 3 of SI.

## Statistical analysis

In order to build an epidemiological model that explicitly takes into account social inequalities, we need to identify which are the main dimensions that interact with age and affect contact patterns the most. To identify these dimensions, we model the expected number of contacts of respondent $i$ using a negative binomial regression[18,51] as defined in Eq. (1):

$$\mu_i = \alpha + \beta_1 \text{age\_group}_i + \beta_2 X_i + \beta_3 \text{age\_group}_i * X_i + \epsilon_i, \quad (1)$$

where $\text{age\_group}_i$ is the age class of $i$; $X_i$ is the variable of interest (e.g., education, income, etc.), $\text{age\_group}_i * X_i$ is the interaction term of the age group and the variable of interest, and $\epsilon_i$ is the error term. Given $\mu_i$, we define $\lambda_i = \exp(\mu_i)$ to be the expected number of contacts for respondent $i$. Then we model the reported number of contacts for respondent $i$, $y_i$, as

$$y_i \sim \text{Neg-Bin}(\lambda_i, \phi), \quad (2)$$

where $\phi \in [1, \infty)$ is a shape parameter that is inversely related to overdispersion: the higher $\phi$ is estimated to be, the closest $y_i$'s distribution is to a Poisson distribution with rate parameter $\lambda_i$.

We build model (1) for each variable of interest ($X$). Particularly, the interaction term between $\text{age\_group}_i$ and the variable of interest allows us to examine whether there are differences in the effect of $X_i$ on the number of contacts in the different age groups. To be able to provide a meaningful description of the interactions, we analyse the average marginal effect (AME)[52–54] of $X_i$ on the number of contacts for different age groups, defined as

$$\text{AME}_{X_i} = \frac{1}{n} \sum_{i=1}^{n} \frac{\partial \mu_i}{\partial \text{age\_group}_i}$$
$$\frac{\partial \mu_i}{\partial \text{age\_group}_i} = \beta_1 + \beta_3 X_i. \quad (3)$$

Working with categorical variables (e.g., education level or employment situation), we can calculate different AMEs for all categories of the categorical variables in each age_group. It means, that, for example, for the interaction of the settlement type (with three categories, out of which one is a reference category) and age (with five categories) on the number of contacts, we have $2 \times 5 = 10$ different AME values: one for each settlement category by each age category. However, as we would like to know the overall effect of the interaction with the given variable, settlement, we have to aggregate these values into one measure. Therefore, we developed the following strategy. For each age group, we examined all the AMEs related to the category of the variable analysed (e.g., all AMEs related to the categories of the settlement type). In each case, we calculated the confidence level[55] at which the confidence interval belonging to the AME of a given category reaches zero. The higher this calculated confidence level is, the more certain we can be that the given variable category has an effect on the number of contacts in the given age group. Equivalently, the estimated effect is significantly different from zero at a smaller type *I* error probability. Out of these confidence levels, we consider the maximum confidence level for each age group as that denotes the highest

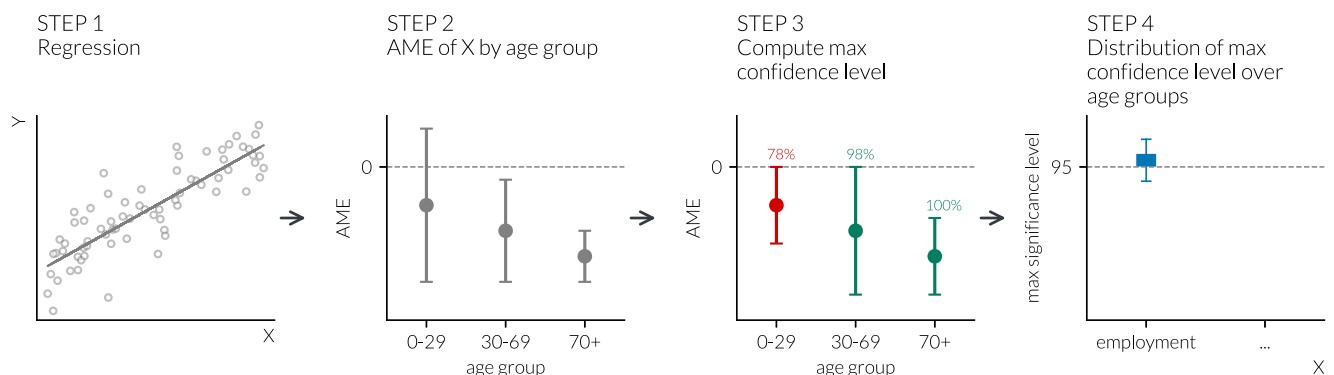

**Fig. 6 | Maximum confidence level computation pipeline.** Pipeline of the statistical analysis to compute the distribution of the *max confidence level* over age groups for each of the variables analysed. The figure is made considering as an illustrative example of the variable *employment*.

confidence level at which the AME of the given variable (e.g., settlement type) reaches zero, i.e. the smallest type *I* error probability[33] at which the estimated effect is significant. Finally, to summarise the results across age groups, we look at its distribution. By following this procedure for each of the variables of interest in the different waves of the pandemic, we are able to rank the variables according to their importance in driving differences in contact patterns additionally to age, in different periods of the COVID-19 pandemic. The pipeline of these analyses is illustrated in Fig. 6.

Following the same methodology, we investigate the dimensions that, in interaction with age, affect the most the probability of getting vaccinated against COVID-19. In this case, we model this probability using a logistic regression model instead of a negative binomial, as the dependent variable was binary and not a count one (see Section 2.2 of SI for further details). Although the data were aggregated to obtain NPIs-homogeneous periods, minor variations in restrictions persist within certain intervals (see SI Fig. 3). Thus, acknowledging the importance of NPIs in shaping human contact during the COVID-19 pandemic, we tested the validity of our results when considering this effect. Additionally, we account for the interplay of vaccination behaviour and contacts, which can be significant in dynamic situations when vaccination strategies are implemented during an epidemic crisis[56]. Specifically, we have included two additional control variables in the statistical model (1): (i) the Oxford Stringency Index, a composite measure quantifying the government's response strictness to the pandemic, and (ii) a dummy variable indicating individual vaccination status (vax$_i$). The results of these robustness analyses are presented in Section 2.1.2 of the SI, along with further robustness analysis where we control for all the other available individual features. Additionally, we investigate the effects of NPIs on vaccination uptake, with results presented in Section 2.2.1 of the SI. All the results of the robustness analysis align with those discussed earlier and lead to the same qualitative conclusions.

## Decoupled contact matrices

Conventionally, to compute age-contact matrices $C_{ij}$ we divide a population into subgroups according to their age and calculate the average number of contacts that individuals in age class $i$ have with individuals in age class $j$[30]. Here, instead, we further stratify individuals from each age class $i$ according to various dimensions, like employment status, settlement or education level.

In detail, we decouple the conventional age contact matrix $C_{ij}$ into $D$ number of matrices, one for each of the subgroups of the dimension that we want to take into account. More precisely, let $\bar{d}$ be the subgroup of the dimension considered and let $\bar{d} \in 1, \ldots, D$. We can write

$$C_{\bar{d}i,j} = CT_{\bar{d}i,j}/N_{\bar{d}i},\qquad(4)$$

where $CT_{\bar{d}i,j}$ is the total number of contacts that individuals of age class $i$ and belonging to subgroup $\bar{d}$ have with individuals in age class $j$, regardless of the subgroup to which the contacted individuals belong; and $N_{\bar{d}i}$ is the total number of individuals in age class $i$ and subgroup $\bar{d}$.

For example, to differentiate between *employed* and *not-employed* individuals, we compute two age contact matrices: $C_{employed,i,j}$ and $C_{not-employed,i,j}$. All these matrices have been corrected for symmetrization as explained in Section 4.3.1 of SI. This framework can be extended to any number of dimensions considered simultaneously, in this case, the length of the $\bar{d}$ vector will correspond to the number of combinations of the levels of the dimensions considered. Note that the available data lack information on the subgroup membership of contacts, recording only the demographic details of survey respondents. Consequently, we opted to decouple the age contact matrices solely along the dimension of the respondent.

## The epidemiological model

In order to investigate the effect of the decoupled contact matrices on the dynamic of infectious disease transmission, we propose a simple mathematical framework as an extension of the conventional age-structured SEIRD compartmental model[34,35].

The conventional SEIRD model is defined as a population where individuals are assigned to five compartments based on their actual state: susceptible (S), exposed (E), infected (I), recovered (R) and dead (D). The model further defines the transition rates of individuals from one compartment to another by incorporating for each age class a given force of infection, which includes the average number of contacts with all the other age classes. The model proposed here extends this definition by taking into account not only the age structure of the contacts in the population but also their differences along a set of other dimensions $\bar{d}$, such as education level, income level and employment situation.

The model can be described by a set of ordinary coupled differential equations as presented in Eq. (5):

$$
\begin{aligned}
\dot{S}_{\bar{d},i} &= -\lambda_{\bar{d},i} S_{\bar{d},i} \\
\dot{E}_{\bar{d},i} &= \lambda_{\bar{d},i} S_{\bar{d},i} - \epsilon E_{\bar{d},i} \\
\dot{I}_{\bar{d},i} &= \epsilon E_{\bar{d},i} - \mu I_{\bar{d},i} \\
\dot{R}_{\bar{d},i} &= \mu(1 - \text{IFR}_i) I_{\bar{d},i} \\
\dot{D}_{\bar{d},i} &= \mu \text{IFR}_i I_{\bar{d},i}.
\end{aligned}
\qquad(5)
$$

Here $i$ indicates the age group of the ego, $j$ indicates the age group of the peer, $\bar{d}$ represents a vector of dimensions to which the ego belongs, $\beta$ is the probability of transmission given a contact, $\epsilon$ is the rate at which individuals become infectious, $\mu$ is the recovery rate, IFR is the infection fatality rate, and $C_{\bar{d}}$ is the age contact matrix corresponding to dimensions $\bar{d}$.

In this system of equations system, we rely on the concept of the force of infection, which is defined as

$$\lambda_{\bar{d},i}(t) = \beta \sum_j \frac{C_{i\bar{d},j}}{N_j} I_j, \tag{6}$$

Further, we rely on the definition of the *infection fatality rate* (IFR$_i$), which is defined as the fraction of infected individuals that died. In order to account for the variability of contacts in our data, for each simulation that we run we use a static decoupled contact matrix that we compute from a bootstrapped sample of our data. See Section 7 of SI for the details on the implementation of the numerical simulations.

### Reporting summary
Further information on research design is available in the Nature Portfolio Reporting Summary linked to this article.

## Data availability
The MASZK survey data cannot be shared openly due to privacy regulations, but they may be available upon request to the corresponding author (karsaim@ceu.edu) after the signature of non-disclosure agreements with the data owner. A sample of data, including contact matrices stratified by age, has been published at https://github.com/adrianamanna/epi_social_inequalities. Data on the Oxford Stringency Index are available at: https://ourworldindata.org/metrics-explained-covid19-stringency-index. Data on the number of vaccinated individuals are available at: https://www.statista.com/statistics/1196109/hungary-number-of-people-vaccinated-against-covid-19/. Data on the total number of deaths in Hungary during the 4th wave are available at: https://kimittud.hu/request/koronavirus_elhunytakra_es_gyogy?nocache=incoming-28514&fbclid=IwAR14PP0DyWIEzlix6mGwNkjHHJmyi8PZLl141vfXeRUzmghjjOqcCBuHx_M#incoming-28514.

## Code availability
We made available a simplified code to (i) derive the decoupled age contact matrix, and (ii) to simulate a *SEIR* model with these matrices together with their visualisation. The code and a sample data set are available at https://github.com/adrianamanna/epi_social_inequalities and https://zenodo.org/doi/10.5281/zenodo.10980134[57].

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

## Acknowledgements

The authors gratefully thank Alessandro Vespignani, Jessica Davis, Eszter Bokányi, Alessia Melegaro and Filippo Trentini for useful discussions. A.M. and M.K. were supported by the Accelnet-Multinet NSF grant. J.K. and M.K. acknowledge funding from the National Laboratory for Health Security (RRF-2.3.1-21-2022-00006). M.K. acknowledges support from the ANR project DATAREDUX (ANR-19-CE46-0008); the SoBigData++ H2020-871042; the EMOMAP CIVICA projects.

## Author contributions

A.M. performed the numerical simulations and data analysis. A.M., J.K. developed the statistical analysis. A.M., M.K. developed the epidemiological model. All authors wrote the first draft of the manuscript. All authors designed the study, interpreted the results, edited and approved the manuscript.

## Funding

## Competing interests

The authors declare no competing interests.
