## [Peer Review File · Nature Communications]

Importance of social inequalities to contact patterns, vaccine uptake, and epidemic dynamicsREVIEWER COMMENTS

Reviewer #1 (Remarks to the Author):

Review of "Social inequalities that matter for contact patterns..."

This paper argues that socio-demographic characteristics of individuals should be incorporated into epidemiological analyses of contact patterns and infectious disease spread. The paper uses survey data collected in Hungary during the Covid-19 pandemic to illustrate this argument with two analyses: first, the authors investigate sociodemographic variation in contact rates and vaccination uptake, with the goal of identifying which characteristics seem most strongly related to contact rates (and to vaccination); second, the results are used to build a model for the dynamics of Covid that incorporates the sociodemographic estimates.

I appreciate the big-picture topic that this paper investigates: I completely agree with the authors that epidemiological modeling has all too often focused only on age and not gone into more socio-economic detail. This study is an impressive effort to go beyond the status quo. That said, the paper read to me like it was not quite a final draft - some editing and rewriting would help make it more readable. I also had a few concerns about the design and analysis (in most cases, these can probably be addressed). I tried to summarize my feedback in point form, separating out the major and minor comments below.

Major comments:

* pg 4 - "note that although all the models have been completed on each pandemic periods [sic]..." - I did not understand why it was appropriate to focus only on the 4th wave. Is the idea that the results are same for all of the waves, and so the authors chose just one to keep the paper as simple as possible? Or are the results different for other waves? If they are different, what justifies focusing on wave 4?

* pg 4 - I think it would be helpful (if possible) to cite some other work related to the 'maximum confidence level.' The concept makes sense to me, but it wasn't immediately obvious that a "higher value indicates that a given variable explains better differences in the number of contacts, given the age of individuals." I find the phrase "explains better" to be unclear - perhaps consider re-wording?

* Fig 2a - Are the differences between average numbers of contacts statistically significant here? (Or, if inferences are being made in a way that means statistical significance is not relevant, please explain that) [the same question could be asked of panels b-d]

* Fig 5 - why compare percent newly infected and the attack rate (and percent newly dead and death rate)? These seem closely related, but not identical (?). I suggest comparing the same quantity if possible, or helping the reader understand why it makes sense not to do that if it's not

* pg 12 - "we find that privileged groups ... are the ones able to better adapt to the epidemiological situation and NPIs by adjusting their number of contacts" - to me, this seems like a somewhat stronger conclusion that the analysis here can support. Although these estimates may suggest that privileged groups reduced their contacts more, I don't think there's any evidence to show that this is because they were able to and other groups wanted to but could not. More generally, I don't think there's any evidence here about ability to adapt - we just see differences in contact patterns (right?). If this is correct, then I suggest removing some of the speculation in this section

* pg 15 - I suggest citing some of the literature to support maximum confidence level and the way it is interpreted here (as an example, the authors do a good job of motivating and citing literature related to the average marginal effect - it would be nice to see that for the maximum confidence level too)

* pg 15 - in the discussion of the decoupled contact matrices, I think it would be helpful to

explicitly say that the data do not have information on the subgroup membership of the contacts - only of the survey respondents. And I suggest incorporating that into the explanation/definition of $C_{\{\bar{d},i,j\}}$. (This is discussed above, but it is important here to help the reader understand how these decoupled contact matrices are defined - they are not what I thought they were the first time I read the paper)

Minor comments:

* pg 2 - "current literature falls short to understand the role of other social, demographics, and economic factors in shaping human behavior that are relevant to the epidemic spreading" - I completely agree

* Fig 1a - is the right-hand axis the average number of contacts, or non-household contacts? If it's the latter, I suggest clarifying the axis label

* Fig 2a - "low-education individuals maintain a lower number of contacts over time with smaller variation" - is the variation also smaller if it is considered as a proportion of the contact level? In other words, instead of the absolute variation, is the relative variation smaller for those with low education?

* Fig 3 - this figure is so small, it's rather hard to read. (The other figures are also on the small side)

* pg 8 - I suggest removing "Convincingly" (it sounds a bit like you are telling the reader what to think)

* pg 9 - "this is because these individuals, having a low number of contacts, are already protected from exposure to the virus thus they gain less from vaccination" - this was interesting, I thought

* pg 10 - "employed people extracted the infection with the highest rate" - I think "extracted" is not the right word here. Maybe 'contracted'?

* pg 11, end of results - I thought this last paragraph was quite hard to follow. Perhaps more structure / organization would help

* pg 12 - "detailed contact diaries ... commonly suffer from recall biase[s] and other limitations" - I suggest adding some citations here

* pg 13 - 'survey response rate was ... expressly higher' - I suggest rewording (I don't think 'expressly' makes sense here)

* pg 13 - 'sampling errors were corrected using iterative proportional post-stratification weights'. A few comments here: (1) post-stratification doesn't require iterative proportional fitting. Perhaps the authors used some kind of raking or calibration? (2) There should be citations to whatever method was used. (3) Post-stratification, raking, etc do not "correct" sampling errors. They can trade bias for variance, with the hope of producing more accurate estimates. I suggest changing the wording to avoid suggesting that they "correct" sampling errors

* Fig 6 - comparing "Step 2" and "Step 3", I could not understand what is happening with the 30-69 age group. It looks to me like the max significance level for that age group should be higher than 95% (assuming that the uncertainty intervals in Step 2 are 95% confidence intervals - though that is not specified anywhere)

Reviewer #2 (Remarks to the Author):

I would like to commend the authors for their efforts in addressing the issue of the impact of socio-economic status on disease spread in the manuscript titled "Social inequalities that matter for contact patterns, vaccination, and the spread of epidemics." The subject matter is undoubtedly of significance, and the research undertaken bears the potential to contribute meaningfully to the academic discourse. The results are noteworthy, but some methodological improvements are direly needed. Although the topic is interesting, there are some crucial aspects that are currently not captured in the current state of the analysis.

First and foremost, the authors aim at modeling vaccination behavior and contact patterns but neglect that both these quantities are driven by a combination of spontaneous (i.e., individual-determined) and nonspontaneous (e.g. due to interventions) behavioral changes. The authors neglect the impact of non spontaneous changes, for example, not including the impact of Non-Pharmaceutical Interventions (NPI) on the number of social contacts. The impact of NPIs on the number of contacts has been confirmed in many, if not all, previous contact studies^{1,2}. This is also true when modeling vaccination behavior, as the authors do not discuss specifics of the vaccination strategy in Hungary (e.g., compulsory vaccination for workers in specific sectors) that may have forced/undetermined vaccination in some individuals. Also, the authors neglect the interplay of vaccination and contacts, which can be quite important in a dynamic situation when vaccination strategies are implemented over an epidemic crisis such as the one of COVID-19³. Including these two effects is crucial to properly model vaccination, behavior, and their interplay.

There are also issues on the methodology regarding the social contact matrices generation that I would like to raise:

When identifying the main determinants of human contact patterns, the authors model each quantity independently. This approach has the evident disadvantage of neglecting potential correlations (for example, between education and employment or settlement and employment, etc), potentially biasing the hierarchy of effects that they measure. Resorting to the usual multivariate regression⁴ would provide a better estimation of the hierarchy of the main determinants. This is even worse when considering that contact matrices are then stratified according to the identified variables, again hiding potential correlations in the stratification procedure.

The authors do not discuss how the stratified contact matrices obey the symmetricity constraint that contacts of individuals in stratum X with individuals in stratum Y is the same of individuals in stratum Y with individuals with stratum X. When the stratum considered is only age, this is simply imposed, but when one is considering more than one dimension (age, education, etc.), this is less trivial. Did the authors consider this issue? The use of unbalanced matrices is known to provide wrong estimates of infectious disease spread⁵.

The authors do not discuss at all the issue of clustering of contacts by socio-economic status. Contact matrices are generated by aggregating the data. This does not account for potential correlations between the variables for which aggregation is performed. The authors should resort to modeling of the contact matrices (in a multivariate framework) in order to properly account for these correlations.

The authors do not discuss how contacts are established between individuals with, e.g., different education. Are individuals with lower education interacting more with individuals with lower education? In other words, what are the assumptions with respect to homophily in the considered determinants of social interaction?

Furthermore, I have some concerns regarding the compartmental model:

The authors only show the simulation error. What about the other uncertainties? For example, how is the variability of contacts in their sample (so far included just as an average) impacting their results on epidemic spread? I think that there is no proper evaluation of uncertainties.

Several parameter values are taken from the literature, however there is no sensitivity analysis to assess the impact of changing these values.

Finally, the impact of socio-economic status on the number of contacts has been previously addressed in Wong et al.² and Quaife et al.⁶. The authors should discuss their results in comparison to these works.

I also have some minor comments on the text, which I list below:

- "Decoupled" to be substituted with aggregated by
- "At the same time, low-educated [...] reflecting their limited social environment and adaption capacities. By looking at the contact dynamics at workplaces, it is evident that only highly educated individuals were able to adapt to the epidemiological situation." These are already considerations to be included in the discussion more than in the results.
- "From an epidemic modelling perspective, [...] via contact matrices that represent a social network at an aggregate level." This is a bit simplified, as several features of the network are not captured by contact matrices (e.g., clustering). I would just state that contact matrices quantify average interactions between population strata.
- "In this case, the interaction with age is particularly important given that the Covid-19 immunization strategy implemented in Hungary followed an age-stratified outreach by prioritizing elderly individuals". Was there any other indication, e.g., compulsory vaccination for specific jobs?
- "It should be noted that our focus is solely on the number of infected individuals. [...] subgroups in modelling, we miss an important determinant, which significantly influence the final outcome of an epidemic". Move to the discussion as this is a limitation of the method.
- "As expected, these curves suggests that group of employed people extracted the infection with the highest rate as compared to not employed others.". I think there is a typo (experienced?)
- "From the simulations we found that although not-employed, low-income and older individuals appeared with the lowest infection rates, they evolved with the highest mortality rate as compared to other groups." Unclear. Is this not to be expected since the age-specific infection fatality rate is higher? If it is an issue of competing effects, discuss it further.
- "Additionally, we find that privileged groups tend to have a higher number of a contact and are the ones able to better adapt to the epidemiological situation and NPIs by adjusting their number of contacts.". I don't think this is only a matter of "ability to better adapt". The authors here measure a combination of spontaneous and non-spontaneous (i.e., induced by NPIs) behavioral change but can not comment on each one separately.
- "More strikingly, we find that privileged groups of the population were more likely to get vaccinated against the Covid-19 virus.". Again here, there is a combination of spontaneous and nonspontaneous effects. Are there jobs for which vaccination is compulsory? Are these jobs providing an income that is randomly distributed, or are these high-paying (or low-paying) jobs? (section Sociodemographic dimension)
- Income: it seems to me that this is "perceived income" more than income. If so, the text should be amended.
- Capital, rural, or urban: how are these defined?

In conclusion, my evaluation of the manuscript suggests that it currently falls short of the quality standards expected for publication in Nature Communications.

References

1. Liu, C. Y. et al. Rapid Review of Social Contact Patterns During the COVID-19 Pandemic. *Epidemiology* 32, 781–791 (2021).
2. Wong, K. L. M. et al. Social contact patterns during the COVID-19 pandemic in 21 European countries – evidence from a two-year study. *BMC Infect. Dis.* 23, 268 (2023).
3. Wambua, J. et al. The influence of COVID-19 risk perception and vaccination status on the number of social contacts across Europe: insights from the CoMix study. <http://medrxiv.org/lookup/doi/10.1101/2022.11.25.22282676> (2022) doi:10.1101/2022.11.25.22282676.
4. Mossong, J. et al. Social Contacts and Mixing Patterns Relevant to the Spread of Infectious Diseases. *PLoS Med.* 5, e74 (2008).
5. Hamilton, M., Knight, J. & Mishra, S. Examining the Influence of Imbalanced Social Contact Matrices in Epidemic Models. *Am. J. Epidemiol.* kwad185 (2023) doi:10.1093/aje/kwad185.
6. Quaipe, M. et al. The impact of COVID-19 control measures on social contacts and transmission in Kenyan informal settlements. *BMC Med.* 18, 316 (2020).

Reviewer #3 (Remarks to the Author):

In the manuscript titled "Social inequalities that matter for contact patterns, vaccination and the spread of epidemics", the authors identify socio-economic determinants responsible for contact behavior and vaccine uptake, making use of data collected from a large-scale survey in Hungary. This analysis is highly informative as it provides a characterization of human behavior relevant to disease spread beyond the usual dimension of age. The impact of social inequalities on the epidemic is further investigated through the use of a data-driven mathematical model that naturally extends the well-known age-structured approach incorporating behavioral differences based also on socio-economic factors (namely employment and income). The authors exploit a rich dataset and a natural extension of a well-known methodology to find valuable insights on the interplay between human behavior and disease dynamics. For this reason, I support the publication of this work. Nevertheless, I attach here some comments for the authors to be addressed before publication.

Figures:

- Figure 1, Figure 2 (a-d): "The white and grey areas delimit the periods that have been aggregated in the analysis: two interim periods (IPs) and four epidemic waves (W)." I would highly suggest to differentiate with color or pattern the interim periods from the epidemic waves. I find the alternating pattern of white and gray a bit misleading. Also, I suggest to indicate relevant dates of implementation of social distancing measures (e.g. start and end of national lockdown) in order to interpret contact trends in relation with the timeline of interventions.
- Figure 1b: what is the horizontal dotted line? Please explain in the caption.
- Figure 1b, 2nd wave: boxplot for income is missing, please explain why.
- Figure 3e-i:
 - typo in C_{idj} , should be C_{dij} following the notation in the text
 - for consistency with the colored lines, I would report the extended SEIR model with continuous lines, and the classical model with dotted lines
- Figure 3a-d: what is the source for the age-stratified population distribution by employment level, education etc? does it refer to the Hungarian population or to the survey population? Please specify

Results section :

- I would ask the authors to clarify how the quantity displayed in Figure 1 was estimated. Is it the crude number of contacts, or adjusted by some variables as usually done in contact studies (e.g. day of the week, population, contact reciprocity)? Also, I would ask to show the associated confidence intervals for each point estimates displayed in the figure (or report them in the SI).
- when referring to the SI, please refer to specific figure numbers or sections to guide the reader.
- description of contact data: data sparsity was mentioned in the last sentence in the results section for some subgroups. I would propose the authors to include a table in the SI with the sample size for each survey wave, stratified by age and socio-economic group. I could not find specific information on the dataset in the manuscript nor in the references cited.
- as acknowledged in the manuscript, the temporal trend of contacts for middle-education individuals at work (Figure 2c) is opposite to what would be expected looking at the other two groups; can the authors provide some insight that can explain this anomalous behavior?
- results in figure 3: it is not clear to me if you used time-varying contact matrices in the epidemic simulation. Do the results on the attack rate still hold, i.e. privileged groups reporting higher attack rate despite the more evident change in behavior and reduction in contacts over time? Or did you use a constant contact matrix?

- validation of epidemic model outcomes: in the discussion the authors briefly mention two studies supporting their results on the unequal COVID-19 impact on different strata in the population. I would suggest to provide more information on the comparison between the results of the manuscript and the results of the studies.

Supplementary Information:

- given the high number of zeros, maybe test a zero-inflated distribution? This is also related to the question above about how you computed the average number of contacts over time. In case of a large number of zeros, the standard mean would not be well-representative unless you model it with a zero-inflated distribution.

- model calibration: it is not clear how the initial recovered/exposed/infected population was distributed across age groups and socio-economic groups.

Detailed response to Reviewers

The Reviewers' comments are reported verbatim.

Reviewer #1 (Remarks to the Author):

Review of "Social inequalities that matter for contact patterns..."

This paper argues that socio-demographic characteristics of individuals should be incorporated into epidemiological analysis of contact patterns and infectious disease spread. The paper uses survey data collected in Hungary during the Covid-19 pandemic to illustrate this argument with two analyses: first, the authors investigate sociodemographic variation in contact rates and vaccination uptake, with the goal of identifying which characteristics seem most strongly related to contact rates (and to vaccination); second, the results are used to build a model for the dynamics of Covid that incorporates the sociodemographic estimates.

I appreciate the big-picture topic that this paper investigates: I completely agree with the authors that epidemiological modeling has all too often focused only on age and not gone into more socio-economic detail. This study is an impressive effort to go beyond the status quo. That said, the paper read to me like it was not quite a final draft - some editing and rewriting would help make it more readable. I also had a few concerns about the design and analysis (in most cases, these can probably be addressed). I tried to summarize my feedback in point form, separating out the major and minor comments below.

Response: We are happy that the Reviewer appreciated the main goals of this work and found our paper an impressive effort to go beyond the status quo. We would like to thank the Reviewer for the detailed feedback, comments, and constructive criticisms. We found them all very valuable and undoubtedly helpful to improve the content and readability of our manuscript. We list below our detailed responses to all of the raised comments, while we modified the manuscript accordingly, with changes highlighted in red.

Major comments:

** pg 4 - "note that although all the models have been completed on each pandemic periods [sic]..." - I did not understand why it was appropriate to focus only on the 4th wave. Is the idea that the results are same for all of the waves, and so the authors chose just one to keep the paper as simple as possible? Or are the results different for other waves? If they are different, what justifies focusing on wave 4?*

Response: We thank the Reviewer for raising this concern. We admit that we were not explaining this seemingly arbitrary choice in enough detail in the previous version of our manuscript. The decision to focus on the 4th wave in the main text of the manuscript is mostly due to the attempt to keep the main text as simple as possible. Presenting the results on all the waves would have been too overwhelming for the Reader. Nevertheless, we include the results of each wave in the SI (see sections 4.3.2 for the matrices, 5 for the vaccination uptake, 7.1 and 7.2 for the results of the epidemic simulations). According to these, the main conclusions are similar if not identical for the different waves. Nevertheless, there were further reasons to concentrate on the 4th wave in the main text and not on others. First of all, this was an epidemic wave during which the non-pharmaceutical interventions were not changing considerably, assuring that our observations would not change due to changing regulations. In addition, for the last part of our analysis (the calibrated scenario) we aimed to select a period where vaccination reached saturation, thus the changing size of the vaccinated population would not bias our conclusions. In the updated manuscript we discuss these reasons for the choice for the 4th wave as the demonstrative period of our study.

** pg 4 - I think it would be helpful (if possible) to cite some other work related to the 'maximum confidence level.' The concept makes sense to me, but it wasn't immediately obvious that a "higher value indicates that a given variable explains better differences in the number of contacts, given the age of individuals." I find the phrase "explains better" to be unclear - perhaps consider re-wording?*

Response: We thank the Reviewer for pointing out this issue, which was not clearly explained in the manuscript. The concept of maximum confidence level was developed by us, in order to aggregate the multiple average marginal effects belonging to the different categories of the interaction terms. We re-worded the interpretation in the main text and also added an extensive description of the procedure and interpretation in the Material and Methods section, to which we referred in the main text as well.

** Fig 2a - Are the differences between average numbers of contacts statistically significant here? (Or, if inferences are being made in a way that means statistical significance is not relevant, please explain that) [the same question could be asked of panels b-d]*

Response: Following up on the comment of the Reviewer, we recognized that the text was not clearly explaining these details. Actually, on these panels we show the average number of contacts as found in the data. However, we acknowledge that, as also Reviewer #2 highlights, our analysis lacked a proper assessment of uncertainties of these values. To include this important information in the revised manuscript, we applied bootstrapping on the corresponding distributions, and computed the IQR values, shown as bounding shaded areas around the average contact values in Fig.2. Since the rest of the paper covers in detail the significant differences among these

quantities with a detailed statistical analysis, we prefer to depict only the trends of these quantities in this figure.

** Fig 5 - why compare percent newly infected and the attack rate (and percent newly dead and death rate)? These seem closely related, but not identical (?). I suggest comparing the same quantity if possible, or helping the reader understand why it makes sense not to do that if it's not*

Response: In Figure 5 our goal is to show the overall outcome of the epidemic as an outcome of our model. We do not aim to compare the percent newly infected with attack rate and the percent newly infected with mortality rate, as these are different outputs of the same model. Additionally, we did not stratify the epidemic curves along all the dimensions simultaneously because it would lead to a high number of subgroups that would have been difficult to show in a unique plot. As we recognize that this was not clear we changed the text in order to clarify it.

** pg 12 - "we find that privileged groups ... are the ones able to better adapt to the epidemiological situation and NPIs by adjusting their number of contacts" - to me, this seems like a somewhat stronger conclusion that the analysis here can support. Although these estimates may suggest that privileged groups reduced their contacts more, I don't think there's any evidence to show that this is because they were able to and other groups wanted to but could not. More generally, I don't think there's any evidence here about the ability to adapt - we just see differences in contact patters (right?). If this is correct, then I suggest removing some of the speculation in this section*

Response: We agree with the Reviewer that the mentioned conclusions were somewhat speculative. Therefore we toned down the corresponding sentences and specified that these conclusions are interpretations of possible (and plausible) mechanisms.

** pg 15 - I suggest citing some of the literature to support maximum confidence level and the way it is interpreted here (as an example, the authors do a good job of motivating and citing literature related to the average marginal effect - it would be nice to see that for the maximum confidence level too)*

We appreciate this observation regarding the clarity of the issue. As the problem has already been raised, we answered it above. To summarize it, the process for the measurement of the maximum confidence level was conceptualized by us to consolidate the various average marginal effects associated with distinct categories of interaction terms. We have revised the interpretation in the main text and incorporated a detailed explanation of the procedure and interpretation in the Material and Methods

section, where we added citation as well. This additional information is also referenced in the main text for comprehensive understanding.

** pg 15 - in the discussion of the decoupled contact matrices, I think it would be helpful to explicitly say that the data do not have information on the subgroup membership of the contacts - only of the survey respondents. And I suggest incorporating that into the explanation/definition of $C_{\{\bar{d},i,j\}}$. (This is discussed above, but it is important here to help the reader understand how these decoupled contact matrices are defined - they are not what I thought they were the first time I read the paper)*

Response: We thank the Reviewer for this comment. We added this information in the *Decoupled contact matrices* section of the revised manuscript.

Minor comments:

** pg 2 - "current literature falls short to understand the role of other social, demographics, and economic factors in shaping human behavior that are relevant to the epidemic spreading" - I completely agree*

Response: We are glad that the Reviewer agrees with our statement.

** Fig 1a - is the right-hand axis the average number of contacts, or non-household contacts? If it's the latter, I suggest clarifying the axis label*

Response: It is the average number of contacts excluding household ones. We now changed the label and specified "excluding household contacts" in the caption of the figure.

** Fig 2a - "low-education individuals maintain a lower number of contacts over time with smaller variation" - is the variation also smaller if it is considered as a proportion of the contact level? In other words, instead of the absolute variation, is the relative variation smaller for those with low education?*

Response: We thank the Reviewer for this interesting comment. In order to answer this question we computed the relative number of contacts, over time by level of education, income, settlement and employment situation. Namely, in each month, the relative number of contacts of each subgroup is computed as the ratio between the average number of contacts for each subgroup and the average of the total number of contacts in the population. The results are consistent with the one previously found. We show these plots in the SI in the section: "4.4.1. Relative contacts variation".

** Fig 3 - this figure is so small, it's rather hard to read. (The other figures are also on the small side)*

Response: We agreed with the Reviewer and we increased the size of all the figures, particularly of Fig3.

** pg 8 - I suggest removing "Convincingly" (it sounds a bit like you are telling the reader what to think)*

Response: We removed the word *Convincingly*.

** pg 9 - "this is because these individuals, having a low number of contacts, are already protected from exposure to the virus thus they gain less from vaccination" - this was interesting, I thought*

Response: We are glad that the Reviewer found our conclusion interesting.

** pg 10 - "employed people extracted the infection with the highest rate" - I think "extracted" is not the right word here. Maybe 'contracted'?*

Response: We corrected this typo with "experienced" .

** pg 11, end of results - I thought this last paragraph was quite hard to follow. Perhaps more structure / organization would help*

Response: We rephrase this paragraph to help better the reader.

** pg 12 - "detailed contact diaries ... commonly suffer from recall biase[s] and other limitations" - I suggest adding some citations here*

Response: Following the suggestion of the Reviewer, we added references to this sentence in the revised manuscript.

** pg 13 - 'survey response rate was ... expressly higher' - I suggest rewording (I don't think 'expressly' makes sense here)*

Response: We substituted the word "expressly" with "notably".

** pg 13 - 'sampling errors were corrected using iterative proportional post-stratification weights'. A few comments here: (1) post-stratification doesn't require iterative proportional fitting. Perhaps the authors used some kind of raking or calibration? (2) There should be citations to whatever method was used. (3) Post-stratification, raking, etc do not "correct" sampling errors. They can trade bias for variance, with the hope of producing more accurate estimates. I suggest changing the wording to avoid suggesting that they "correct" sampling errors*

Response: Thank you for pointing out this issue. In the Data description section we re-phrased the text to make it more correct and added a citation for the iterative proportional fitting.

** Fig 6 - comparing "Step 2" and "Step 3", I could not understand what is happening with the 30-69 age group. It looks to me like the max significance level for that age group should be higher than 95% (assuming that the uncertainty intervals in Step 2 are 95% confidence intervals - though that is not specified anywhere).*

Response: We thank the Reviewer for this comment. There was indeed an error in Figure 6, that we corrected in the revised manuscript. We also corrected the title of panel 3 and 4 to be consistent with Figure 1.

Reviewer #2 (Remarks to the Author):

I would like to commend the authors for their efforts in addressing the issue of the impact of socio-economic status on disease spread in the manuscript titled "Social inequalities that matter for contact patterns, vaccination, and the spread of epidemics." The subject matter is undoubtedly of significance, and the research undertaken bears the potential to contribute meaningfully to the academic discourse. The results are noteworthy, but some methodological improvements are direly needed. Although the topic is interesting, there are some crucial aspects that are currently not captured in the current state of the analysis.

Response: We are happy that the Reviewer finds that our research can potentially contribute meaningfully to the academic discourse. We express our gratitude for providing the valuable feedback, insightful comments, and constructive criticisms. These inputs were immensely beneficial and unquestionably contributed to enhancing the quality of the manuscript. We list below our detailed responses to all of the raised comments, while we modified the manuscript accordingly, with changes highlighted in red.

Comments:

First and foremost, the authors aim at modeling vaccination behavior and contact patterns but neglect that both these quantities are driven by a combination of spontaneous (i.e., individual-determined) and nonspontaneous (e.g. due to interventions) behavioral changes. The authors neglect the impact of non spontaneous changes, for example, not including the impact of Non-Pharmaceutical Interventions (NPI) on the number of social contacts. The impact of NPIs on the number of contacts has been confirmed in many, if not all, previous contact studies [1,2].

Response: We thank the Reviewer for this insightful comment. First we would like to clarify that the intention of the statistical model is not to model the total number of contacts or the vaccination uptake in order to find the variables that can predict those. Rather, our aim is to identify those socio-demographic variables - in interaction with age -, along which the number of contacts and vaccination uptake differ the most. That is, given that all individuals are exposed to the same exogenous situation, our goal is to identify along which dimension they are varying the most.

We agree with the Reviewer that NPIs strongly affect the number of contacts and we admit that we were not enough verbal about it in the text. To minimize the effects of NPIs and changing epidemiological situations, we chose to present our observations on the period of the 4th wave, during which interventions were not changed significantly (see the dynamics of the overall stringency index see Supplementary Figure 3). Meanwhile, during this period the vaccination rate has already reached its

stationary value, thus neither that affected our observations considerably. This way, we could assume that the population was exposed to similar and stable exogenous factors during this period.

However, to further account for the influence of NPIs on the observed differences, we decided to incorporate NPIs in our analysis presented in the revised manuscript. We replicated the statistical analyses with an additional control variable: *the Oxford Stringency Index*. This index is a composite measure that aims to quantify the strictness of the government's response to the Covid-19 pandemic. It is computed as a composite measure of nine of nine metrics. We summarize the results of the additional statistical analysis in Section 2.1.2 of the SI, which show similar results driving us to the same qualitative conclusions.

This is also true when modeling vaccination behavior, as the authors do not discuss specifics of the vaccination strategy in Hungary (e.g., compulsory vaccination for workers in specific sectors) that may have forced/undetermined vaccination in some individuals.

Response: We thank the Reviewer for this comment. As also mentioned above, in the mentioned statistical analyses the goal was to detect those variables, which have significant effect on vaccination uptake in interaction with age - and not to model and unfold the underlying mechanisms through which these interaction effects prevail. The reason to support the decision of our statistical model were the following (i) all the population was subject to the same kind of vaccination strategy, (ii) although, the vaccination strategy in Hungary prioritize some type of workers in the very beginning of the vaccination period, the main determinant was age, which we carefully consider in our model by adding it as a control variable and as interaction term. However, to further investigate the effects of NPIs on vaccination uptake we repeated the analysis adding *the Oxford Stringency Index* as a control variable. We show the results in section 2.2.1. of the SI.

Also, the authors neglect the interplay of vaccination and contacts, which can be quite important in a dynamic situation when vaccination strategies are implemented over an epidemic crisis such as the one of COVID-19 [3]. Including these two effects is crucial to properly model vaccination, behavior, and their interplay.

Response: We carefully considered this comment of the Reviewer and, in order to account for the interplay of contact patterns and vaccination behavior we decided to add an additional control variable to the statistical models, where the dependent variable is the number of contacts. Namely, we added a dummy variable indicating if an individual was vaccinated or not against COVID-19 to control for the effect of vaccination status. We discuss the results in section 2.1.2. of the SI.

There are also issues on the methodology regarding the social contact matrices generation that I would like to raise:

When identifying the main determinants of human contact patterns, the authors model each quantity independently. This approach has the evident disadvantage of neglecting potential correlations (for example, between education and employment or settlement and employment, etc), potentially biasing the hierarchy of effects that they measure. Resorting to the usual multivariate regression [4] would provide a better estimation of the hierarchy of the main determinants. This is even worse when considering that contact matrices are then stratified according to the identified variables, again hiding potential correlations in the stratification procedure.

Response: As highlighted also in the previous answers, the goal of these analyses was to detect those variables, which have significant effect on the number of contacts in interaction with age - and not to model and unfold the underlying mechanisms through which these interaction effects prevail. Nevertheless, we thank the Reviewer for highlighting the problem of potential correlation between the social-demographic variables. To control for potential correlations and to check the robustness of the results we conducted the suggested multivariate regressions. In these regressions, we added all other social-demographic variables as control variables into the model, as well as the Oxford Stringency Index (accounting for the strictness of the NPIs) and the vaccination status. We extended the SI with the description of the procedure and the presentation of the results of the new models (Fig. 6 and Fig 7 in the SI). As we concluded there, the results closely align with those presented earlier in the main text, reinforcing the robustness of our qualitative conclusions.

The authors do not discuss how the stratified contact matrices obey the symmetricity constraint that contacts of individuals in stratum X with individuals in stratum Y is the same of individuals in stratum Y with individuals with stratum X. When the stratum considered is only age, this is simply imposed, but when one is considering more than one dimension (age, education, etc.), this is less trivial. Did the authors consider this issue? The use of unbalanced matrices is known to provide wrong estimates of infectious disease spread.

Response: We thank the Reviewer for this comment, it is indeed a crucial issue. As mentioned by the Reviewer when it comes to decoupled-age contact matrices the condition of reciprocity is less trivial and it is not straightforward to impose the condition of symmetricity. In the first draft of the manuscript we kept matrices as computed from the raw data. However, in the revised manuscript we considered this issue by explanation and recalculation. We added a section (4.3.1) in the SI called “Symmetrization of the decoupled contact matrices” in which we explain how we symmetrize the decoupled age contact matrices. Meanwhile, all the analyses have

been recalculated with the symmetrized decoupled age-contact matrices and the results haven't changed with respect to the previous analysis.

The authors do not discuss at all the issue of clustering of contacts by socio-economic status. Contact matrices are generated by aggregating the data. This does not account for potential correlations between the variables for which aggregation is performed. The authors should resort to modeling of the contact matrices (in a multivariate framework) in order to properly account for these correlations.

Response: We thank the Reviewer for posing this very interesting question. We strongly agree with this concern, and this brought us to another work of us: "Generalized contact matrices for epidemic modeling" (<https://doi.org/10.48550/arXiv.2306.17250>) in which we consider a fully double-stratified contact matrix (age and another variable), that allows to identify also assortative (clustering) and disassortative patterns along the second (or further) dimensions taken into account. Although this is an extremely important point, in the actual work, because of the missing information on the contactee other than age recorded during the whole study, we could only stratify matrices along the dimensions of the ego. We acknowledge this limitation in our work in the Discussion session of the revised manuscript.

The authors do not discuss how contacts are established between individuals with, e.g., different education. Are individuals with lower education interacting more with individuals with lower education? In other words, what are the assumptions with respect to homophily in the considered determinants of social interaction?

Response: As mentioned above, due to the lack of information on the contactee level of education, income and any other variable but age, we can only stratify the matrix along the characteristic of the respondent. Thus, we did not make any assumption on the assortativity pattern along the second dimension, as in that dimension we only model the contact by age. This is highlighted in the discussion section.

Furthermore, I have some concerns regarding the compartmental model:

The authors only show the simulation error. What about the other uncertainties? For example, how is the variability of contacts in their sample (so far included just as an average) impacting their results on epidemic spread?

Response: We thank the Reviewer for raising this problem. To address this comment and to assess the uncertainties better, we now performed a bootstrapped technique that allows us to better capture the uncertainties in our sample, and particularly in the number of contacts and of vaccination uptake. All the results are now shown considering the bootstrapped IQR. In addition we add section 3 ("Bootstrapping") in the SI in which we explain the technique that we implemented.

I think that there is no proper evaluation of uncertainties.

Several parameter values are taken from the literature, however there is no sensitivity analysis to assess the impact of changing these values.

Response: Following the suggestion of the Reviewer, to assess the uncertainties related to the parameter values, we performed a sensitivity analysis and we present the results in Sections 7.1.1 and 7.2.2 of the SI. These results show expected dependencies on the selected parameters. Also note, even if we make a parameter fit on the Hungarian observations, our study does not aim to provide any forecasting result about the epidemic outcome. We aim only to demonstrate the non-trivial dependence of the epidemic outcome on the studies dimensions.

Finally, the impact of socio-economic status on the number of contacts has been previously addressed in Wong et al.2 and Quaife et al.6. The authors should discuss their results in comparison to these works.

Response: We thank the Reviewer for bringing to our attention the existence of these papers. As we recognize that these provide insightful results on the impact of NPIs on contact patterns by socio-demographic and economic variables, we decided to cite Wong at all in the Introduction and Quaife et al in the discussion section and briefly comment on them.

I also have some minor comments on the text, which I list below:

- “Decoupled” to be substituted with aggregated by

Response: We carefully evaluated this comment of the Reviewer, however, as the narrative of the paper considers the age-contact matrices as the aggregated one, we preferred to refer to the proposed matrices as “decoupled matrices”.

- “At the same time, low-educated [...] reflecting their limited social environment and adaption capacities. By looking at the contact dynamics at workplaces, it is evident that only highly educated individuals were able to adapt to the epidemiological situation.” These are already considerations to be included in the discussion more than in the results.

Response: We removed any form of speculation from the results section, and this text and additional comments on it have been moved to the Discussion section.

-“From an epidemic modelling perspective, [...] via contact matrices that represent a social network at an aggregate level.” This is a bit simplified, as several features of the network are not captured by contact matrices (e.g., clustering). I would just state that contact matrices quantify average interactions between population strata.

Response: We substituted the sentence “.. that represents a social network at an aggregate level.” with: “ .. that quantify the average number of interactions between population strata”

-“In this case, the interaction with age is particularly important given that the Covid-19 immunization strategy implemented in Hungary followed an age-stratified outreach by prioritizing elderly individuals”. Was there any other indication, e.g., compulsory vaccination for specific jobs?

Response: As mentioned above, although the vaccination strategy in Hungary prioritized some type of workers for a limited time, the main determinant was age.

-“It should be noted that our focus is solely on the number of infected individuals. [...] subgroups in modeling, we miss an important determinant, which significantly influences the final outcome of an epidemic”. Move to the discussion as this is a limitation of the method.

Response: We agreed with the Reviewer and we moved these sentences to the Discussion section.

- “As expected, these curves suggest that a group of employed people extracted the infection with the highest rate as compared to not employed others.”. I think there is a typo (experienced?)

Response: We thank the Reviewer for spotting this typo. We corrected it.

-“From the simulations we found that although not-employed, low-income and older individuals appeared with the lowest infection rates, they evolved with the highest mortality rate as compared to other groups.” Unclear. Is this not to be expected since the age-specific infection fatality rate is higher? If it is an issue of competing effects, discuss it further.

Response: We agreed with the Reviewer and we clarified this in the text.

- *“Additionally, we find that privileged groups tend to have a higher number of contacts and they are the ones able to better adapt to the epidemiological situation and NPIs by adjusting their number of contacts.” I don’t think this is only a matter of “ability to better adapt”. The authors here measure a combination of spontaneous and non-spontaneous (i.e., induced by NPIs) behavioral change but can not comment on each one separately.*

Response: We agreed with the Reviewer and we rephrase this paragraph.

- *“More strikingly, we find that privileged groups of the population were more likely to get vaccinated against the Covid-19 virus.”. Again here, there is a combination of spontaneous and nonspontaneous effects. Are there jobs for which vaccination is compulsory? Are these jobs providing an income that is randomly distributed, or are these high-paying (or low-paying) jobs?*

Response: We thank the Reviewer for this comment as mentioned above, although the vaccination strategy in Hungary prioritizes some types of workers for a limited time, the main determinant was age. The prioritized sectors provided medical and other essential services and employed people with a broad range of income. Nevertheless, since their vaccination was prioritized only in the very beginning of the vaccination period, the number of people receiving vaccination in this way was limited. We modified the text to be more precise.

(section Sociodemographic dimension)

- *Income: it seems to me that this is “perceived income” more than income. If so, the text should be amended.*

- *Capital, rural, or urban: how are these defined?*

Response: We are grateful for this comment of the Reviewer, as indeed we were not entirely precise in explaining the income indicator we employ in our analysis. We now specify both in the main and in the SI that by income we actually refer to “perceived income” and we added the definition of Capital, rural and urban in the Supplementary information.

In conclusion, my evaluation of the manuscript suggests that it currently falls short of the quality standards expected for publication in Nature Communications.

References

- 1. Liu, C. Y. et al. Rapid Review of Social Contact Patterns During the COVID-19 Pandemic. Epidemiology 32, 781–791 (2021).*
- 2. Wong, K. L. M. et al. Social contact patterns during the COVID-19 pandemic in 21 European countries – evidence from a two-year study. BMC Infect. Dis. 23, 268 (2023).*

3. Wambua, J. et al. *The influence of COVID-19 risk perception and vaccination status on the number of social contacts across Europe: insights from the CoMix study.* <http://medrxiv.org/lookup/doi/10.1101/2022.11.25.22282676> (2022)
doi:10.1101/2022.11.25.22282676.
4. Mossong, J. et al. *Social Contacts and Mixing Patterns Relevant to the Spread of Infectious Diseases.* *PLoS Med.* 5, e74 (2008).
5. Hamilton, M., Knight, J. & Mishra, S. *Examining the Influence of Imbalanced Social Contact Matrices in Epidemic Models.* *Am. J. Epidemiol.* kwad185 (2023)
doi:10.1093/aje/kwad185.
6. Quaife, M. et al. *The impact of COVID-19 control measures on social contacts and transmission in Kenyan informal settlements.* *BMC Med.* 18, 316 (2020).

Reviewer #3 (Remarks to the Author):

In the manuscript titled “Social inequalities that matter for contact patterns, vaccination and the spread of epidemics”, the authors identify socio-economic determinants responsible for contact behavior and vaccine uptake, making use of data collected from a large-scale survey in Hungary. This analysis is highly informative as it provides a characterization of human behavior relevant to disease spread beyond the usual dimension of age. The impact of social inequalities on the epidemic is further investigated through the use of a data-driven mathematical model that naturally extends the well-known age-structured approach incorporating behavioral differences based also on socio-economic factors (namely employment and income). The authors exploit a rich dataset and a natural extension of a well-known methodology to find valuable insights on the interplay between human behavior and disease dynamics. For this reason, I support the publication of this work. Nevertheless, I attach here some comments for the authors to be addressed before publication.

Response: We are glad that the Reviewer supports our work for publication. At the same time, we express our gratitude for the valuable feedback, comments, and constructive criticisms. These inputs were beneficial and undoubtedly contributed to enhancing the quality of the manuscript.

Figures:

- Figure 1, Figure 2 (a-d): “The white and grey areas delimit the periods that have been aggregated in the analysis: two interim periods (IPs) and four epidemic waves (W).” I would highly suggest to differentiate with color or pattern the interim periods from the epidemic waves. I find the alternating pattern of white and gray a bit misleading. Also, I suggest to indicate relevant dates of implementation of social distancing measures

(e.g. start and end of national lockdown) in order to interpret contact trends in relation with the timeline of interventions.

Response: Following the comment of the Reviewer, we changed the delimitation to better indicate the epidemic waves and interim periods. In the revised manuscript interim periods appear as white, while waves are indicated by dashed gray periods. In addition, we indicated some of the most significant events during the COVID-19 pandemic in Hungary. These are now presented in Figure 1a with dotted vertical lines and a brief description of the event.

- Figure 1b: what is the horizontal dotted line? Please explain in the caption.

Response: We thank the Reviewer for this comment and we admit that this information was actually missing from the text. The horizontal dotted line is placed at 95% and it represents the confidence level at which the AME is considered to be statistically significant. We added this information in the caption.

- Figure 1b, 2nd wave: boxplot for income is missing, please explain why.

Response: Indeed, this information was actually missing from the text. The box plot was missing because income has been collected since the 9th data collection wave only. We added this information in the caption in the main text and in subsection 1.1 in the Supplementary Information.

- Figure 3e-i:

- typo in C_{idj} , should be C_{dij} following the notation in the text

- for consistency with the colored lines, I would report the extended SEIR model with continuous lines, and the classical model with dotted lines

Response: We corrected this typo and we now report the extended SEIR model with continuous lines and the classical model with dotted lines.

- Figure 3a-d: what is the source for the age-stratified population distribution by employment level, education etc? Does it refer to the Hungarian population or to the survey population? Please specify

Response: We thank the Reviewer for this comment and we recognize that this information was actually missing from the text. We refer to the survey population, we now specify this in the caption of Figure 3. However, as the survey was conducted on a representative sample of the Hungarian population

[\[https://www.nature.com/articles/s41598-022-07488-7\]](https://www.nature.com/articles/s41598-022-07488-7), and we are confident that it reproduces faithfully the true distribution of the population in Hungary.

Results section :

- I would ask the authors to clarify how the quantity displayed in Figure 1 was estimated. Is it the crude number of contacts, or adjusted by some variables as usually done in contact studies (e.g. day of the week, population, contact reciprocity)? Also, I would ask to show the associated confidence intervals for each point estimates displayed in the figure (or report them in the SI).

Response: We thank the Reviewer for this comment. The quantity in Figure 1 is the average number of contacts that occurred the day before the interview reported by individuals. This quantity is computed by accounting for the weight of each individual in the survey population and contains all contacts outside of home (including contexts of community, school and work). In order to assess the uncertainty concerning our estimates we applied a bootstrap technique. Now all the results are shown as the median of 1000 simulation and it's Interquartile range (IQR).

- when referring to the SI, please refer to specific figure numbers or sections to guide the reader.

Response: We added more specific references in the main text regarding the SI.

- description of contact data: data sparsity was mentioned in the last sentence in the results section for some subgroups. I would propose the authors include a table in the SI with the sample size for each survey wave, stratified by age and socio-economic group. I could not find specific information on the dataset in the manuscript nor in the references cited.

Response: We agree with the Reviewer that clear information regarding this aspect of the data was missing from the previous version of the manuscript. In the Supplementary Information we added section 1.2 including eight tables, one for each of the variables of interest (*education level, income, settlement, employment situation, acute disease, chronic disease and smoking*).

- as acknowledged in the manuscript, the temporal trend of contacts for middle-education individuals at work (Figure 2c) is opposite to what would be expected looking at the other two groups; can the authors provide some insight that can explain this anomalous behavior?

Response: We thank the Reviewer for pointing out this. We further analyzed the temporal trend of individuals in each of the education groups. We find out that the “opposite trend” was mostly due to individuals with the highest education level of vocational schools, which in Hungary is often linked to interactive work. Thus we decided to modify the classification of the education variable to be able to separate this group, which is special from the perspective of their contact-intensive work, even during a pandemic. This new classification, that now contains 4 levels instead of 3, can illustrate the social processes in a better way and shows the different trends in a more understandable manner. We now present all the results according to this new classification.

- results in figure 3: it is not clear to me if you used time-varying contact matrices in the epidemic simulation. Do the results on the attack rate still hold, i.e. privileged groups reporting higher attack rate despite the more evident change in behavior and reduction in contacts over time? Or did you use a constant contact matrix?

Response: For each of the periods that we consider we use a different set of static decoupled contact matrices. Particularly we now clarify this in Section 7 of the Supplementary Information. We employed daily varying contact matrices in another work already (Bokányi, E., et al., Sci. Rep., 13(1), 21452 (2023)) that were inferred from a non-representative online survey collected in parallel. Although these daily contact matrices were periodically corrected by the actual representative data, they appeared less representative thus we decided not to use them in the actual study.

- validation of epidemic model outcomes: in the discussion the authors briefly mention two studies supporting their results on the unequal COVID-19 impact on different strata in the population. I would suggest to provide more information on the comparison between the results of the manuscript and the results of the studies.

Response: Following the Reviewer’s suggestion, we added more details about the papers we are referring to.

Supplementary Information:

- given the high number of zeros, maybe test a zero-inflated distribution? This is also related to the question above about how you computed the average number of contacts over time. In case of a large number of zeros, the standard mean would not be well-representative unless you model it with a zero-inflated distribution.

Response: We thank the Reviewer for this comment. The means of the contact numbers were calculated from the empirical distributions of the number of contacts,

and not from the models. However, as we are aware of the high number of zeros we decided that in the modeling part, we perform additional statistical analysis on (i) on the number of contacts equal to 0 vs different from 0 (logit), (ii) only on the contact larger than 0, which aligns with the logic of the zero-inflated models. Due to data sparsity we could not run zero inflated regression models, but as the raised problem was important for us as well, we decided to apply this solution. We detailed this analysis in the SI (2.1.1 Accounting for the high number of zeros).

- model calibration: it is not clear how the initial recovered/exposed/infected population was distributed across age groups and socio-economic groups.

Response: We are thankful for this comment as indeed this information was missing from the text. We distributed recovered/exposed/infected individuals in age groups and socio-economic groups proportionally to the population distribution. We clarified this detail in the SI in the “Model calibration section”

REVIEWER COMMENTS

Reviewer #1 (Remarks to the Author):

I thank the authors for their response memo. They have made a good effort to address the concerns that I raised in my previous review.

There is only one issue I was not completely persuaded has been resolved:

* "For having more accurate estimates, we applied post stratification weighting on the data. The weights were calculated using iterative proportional fitting." - I still think this is a little unclear, and potentially a little misleading. Post-stratification is just a matter of division - there is no 'fitting' needed. If the approach used was not post-stratification, but calibration (or, as a special case of calibration, raking) then something like iterative proportional fitting would be needed. My suggestions are to:

- make it clearer that we do not know that this improves accuracy, but this procedure is performed with the hope of improving accuracy
- tell us how the weights were calculated in more detail (probably in the appendix), including the software that was used and the name of the method that was used (again, if IPF was needed, then I don't think it was post-stratification - but whatever it was, more clarity would be helpful)

Reviewer #2 (Remarks to the Author):

I thank the author for taking the time to address and discuss all the issues the other reviewers and I raised. I commend their good work, which improved the quality of the analysis. However, I think that some of the discussions that are part of the point-per-point response and that generated several additional analyses should be discussed better in the paper. This would provide the readers the additional insights that this review process generated and would highlight some aspects of the work that are of crucial importance.

In particular I am referring to:

- Impact of NPIs: The authors should summarize the approach and the results of Section 2.1.2 of the supporting information. There is no reference to the Oxford Stringency Index in the main paper, how this could be used to quantify the extent of NPIs and how this has been used in the study to control for varying levels of NPIs.
- Interplay of vaccination and contacts: the authors should acknowledge the interplay between the two, as this is something that they do not include in their work and that the readers should be aware of, in order to better interpret the results and put them into perspective. Given the available evidence of this effect, this should be even considered as one of the main limitations of this study.

I think that, once these two aspects are better integrated in the main paper, the manuscript is ready for submission. The authors already did a great job in providing further evidence to address the comments of the reviewers.

Reviewer #3 (Remarks to the Author):

I would like to commend the authors for their effort in revising the manuscript, especially addressing the reviewers' comments about assessing uncertainty, robustness and running sensitivity analyses. These have certainly improved the strength of the paper.

I have some final points to be addressed upon publication.

- 1) In response to the authors' reply to my comment about model calibration: "We distributed recovered/exposed/infected individuals in age groups and socio-economic groups proportionally to

the population distribution" I think that the authors need to acknowledge that this homogeneous initial condition is unlikely to be true because different age and socio-economics groups have been heterogeneously affected, precisely because of the different contact patters, differences in exposure risk and so on. Maybe you should acknowledge that in the text as a limitation.

2) I highly suggest a careful revision of the writing, because I found many typos in the text, here are some examples: data prepossessing, perceived income, results are sown, house-hold, those where associated, etc.

Detailed response to Reviewers

Reviewer #1 (Remarks to the Author):

We are thankful for the Reviewer to acknowledge our efforts to address her/his comments, that helped us considerably to improve our manuscript. Below we summarize our detailed answer to the only remaining comment of the Reviewer.

Comment:

I thank the authors for their response memo. They have made a good effort to address the concerns that I raised in my previous review.

There is only one issue I was not completely persuaded has been resolved:

* "For having more accurate estimates, we applied post stratification weighting on the data. The weights were calculated using iterative proportional fitting." - I still think this is a little unclear, and potentially a little misleading. Post-stratification is just a matter of division - there is no 'fitting' needed. If the approach used was not post-stratification, but calibration (or, as a special case of calibration, raking) then something like iterative proportional fitting would be needed. My suggestions are to:

- make it clearer that we do not know that this improves accuracy, but this procedure is performed with the hope of improving accuracy
- tell us how the weights were calculated in more detail (probably in the appendix), including the software that was used and the name of the method that was used (again, if IPF was needed, then I don't think it was post-stratification - but whatever it was, more clarity would be helpful)

Response: *Thank you for pointing out that this issue is still unclear. Your comment helped us to understand why the quoted part was vague. We have clarified the text based on your suggestions. Since the applied solution (raking) is not a conventional post-stratification procedure, we have removed the term 'post-stratification' from the text in order to avoid any possibility of misunderstanding.*

In the main text, we have clarified the objective of the weighting and the type of solution used (raking) to calculate the weights. We have added a note that more details about the weighting are available in Section 1.3 of the SI. In Section 1.3 of the SI, we have described the calculation in more detail, including the software and function used for the creation of weights. We hope that the concept of weighting is clear in the new version.

Reviewer #2 (Remarks to the Author):

We are thankful for the Reviewer to acknowledge our efforts to address her/his comments, that helped us considerably to improve our manuscript. Below we summarize our detailed answer to the remaining comments of the Reviewer.

Comments:

I thank the author for taking the time to address and discuss all the issues the other reviewers and I raised. I commend their good work, which improved the quality of the analysis. However, I think that some of the discussions that are part of the point-per-point response and that generated several additional analyses should be discussed better in the paper. This would provide the readers the additional insights that this review process generated and would highlight some aspects of the work that are of crucial importance.

In particular I am referring to:

- Impact of NPIs: The authors should summarize the approach and the results of Section 2.1.2 of the supporting information. There is no reference to the Oxford Stringency Index in the main paper, how this could be used to quantify the extent of NPIs and how this has been used in the study to control for varying levels of NPIs.
- Interplay of vaccination and contacts: the authors should acknowledge the interplay between the two, as this is something that they do not include in their work and that the readers should be aware of, in order to better interpret the results and put them into perspective. Given the available evidence of this effect, this should be even considered as one of the main limitations of this study.

I think that, once these two aspects are better integrated in the main paper, the manuscript is ready for submission. The authors already did a great job in providing further evidence to address the comments of the reviewers.

Responses: *We thank the reviewer for highlighting that these two additional analyses have not been properly discussed in the main text. We now refer to the analysis concerning the Oxford Index and the NPI in the sub section Statistical analysis of the section Materials and Methods, while presenting the detailed analysis in the Supplementary Information.*

In addition, we briefly discuss the interplay between vaccination and contact as a limitation of our study in the Discussion Section of the main text.

Reviewer #3 (Remarks to the Author):

We are thankful for the Reviewer to acknowledge our efforts to address her/his comments, that helped us considerably to improve our manuscript. Below we summarize our detailed answer to the remaining comment of the Reviewer.

Comments:

I would like to commend the authors for their effort in revising the manuscript, especially addressing the reviewers' comments about assessing uncertainty, robustness and running sensitivity analyses. These have certainly improved the strength of the paper.

I have some final points to be addressed upon publication.

1) In response to the authors' reply to my comment about model calibration: "We distributed recovered/exposed/infected individuals in age groups and socio-economic groups proportionally to the population distribution" I think that the authors need to acknowledge that this homogeneous initial condition is unlikely to be true because different age and socio-economics groups have been heterogeneously affected, precisely because of the different contact patters, differences in exposure risk and so on. Maybe you should acknowledge that in the text as a limitation.

2) I highly suggest a careful revision of the writing, because I found many typos in the text, here are some examples: data prepossessing, perceived income, results are sown, house-hold, those where associated, etc.

Response: *Following these comments of the Reviewer, we now include the initialization of the calibration model as a limitation in the Discussion Section of the main manuscript.*

Also, we re-read the text carefully and corrected all detected typos.